# MAPP unravels frequent co-regulation of splicing and polyadenylation by RNA-binding proteins and their dysregulation in cancer

Maciej Bak[1,2], Erik van Nimwegen [1,2], Ian U. Kouzel[3], Tamer Gur[3], Ralf Schmidt[1,2], Mihaela Zavolan [1,2] & Andreas J. Gruber [3] ✉

Maturation of eukaryotic pre-mRNAs via splicing and polyadenylation is modulated across cell types and conditions by a variety of RNA-binding proteins (RBPs). Although there exist over 1,500 RBPs in human cells, their binding motifs and functions still remain to be elucidated, especially in the complex environment of tissues and in the context of diseases. To overcome the lack of methods for the systematic and automated detection of sequence motif-guided pre-mRNA processing regulation from RNA sequencing (RNA-Seq) data we have developed MAPP (Motif Activity on Pre-mRNA Processing). Applying MAPP to RBP knock-down experiments reveals that many RBPs regulate both splicing and polyadenylation of nascent transcripts by acting on similar sequence motifs. MAPP not only infers these sequence motifs, but also unravels the position-dependent impact of the RBPs on pre-mRNA processing. Interestingly, all investigated RBPs that act on both splicing and 3' end processing exhibit a consistently repressive or activating effect on both processes, providing a first glimpse on the underlying mechanism. Applying MAPP to normal and malignant brain tissue samples unveils that the motifs bound by the PTBP1 and RBFOX RBPs coordinately drive the oncogenic splicing program active in glioblastomas demonstrating that MAPP paves the way for characterizing pre-mRNA processing regulators under physiological and pathological conditions.

Splicing and 3' end processing of nascent RNAs are crucial steps in the maturation of eukaryotic precursor messenger RNA (pre-mRNA), also responsible for transcriptome diversification through the generation of transcript isoforms. Both processes are modulated by various RNA-binding proteins (RBPs), whose expression varies across tissues. To date, a few dozen regulators have been described to modulate splicing[1,2], whereas only a handful were reported to impact both splicing and 3' end processing. The Poly(rC) Binding Protein 1 (PCBP1) RBP

is a known splicing regulator[3], which has also been reported to regulate the cleavage and polyadenylation (poly(A)) of transcript 3' ends by binding to C-rich sequences that are located in close proximity to poly(A) sites[4]. Further, in previous studies we have shown that the well-known splicing factors HNRNPC (Heterogeneous Nuclear Ribonucleoprotein C)[5] and PTBP1 (Polypyrimidine Tract Binding Protein 1)[6] regulate alternative cleavage and polyadenylation (APA) by binding to sequence motifs that are located within −200 to +100 and −25 to +75

[1]Swiss Institute of Bioinformatics, 1015 Lausanne, Switzerland. [2]Biozentrum, University of Basel, 4056 Basel, Switzerland. [3]Department of Biology, University of Konstanz, D-78464 Konstanz, Germany. ✉e-mail: gruber@uni-konstanz.de

nucleotides (nt), respectively, relative to the regulated poly(A) sites. ELAVL1 (ELAV Like RNA-binding Protein 1) is another RBP that was reported to impact splicing[7] and polyadenylation[8]. Also the TAR DNA Binding Protein (TARDBP) is known to act as a regulator of alternative splicing (AS)[9] and APA[10]. While these examples indicate that RBPs coordinately regulate splicing and 3′ end processing, the sparse characterization of binding specificities for the more than 1500 RBPs encoded in the human genome[11] has limited these studies. To circumvent this problem we have developed MAPP (Motif Activity on Pre-mRNA Processing). MAPP enables the identification of RBP-specific sequence motifs that can explain global patterns of both alternative splicing and alternative polyadenylation events quantified from standard RNA sequencing (RNA-Seq) experiments. MAPP further unravels the type of regulation (repressive or activating) as well as the binding site position dependency and, by charting RBP impact maps, MAPP provides a panoramic view of the regulation of alternative splicing and polyadenylation by specific RBPs. We have benchmarked MAPP using datasets in which RBPs with well-characterized impact on splicing and/ or 3′ end processing have been overexpressed or depleted by siRNA-mediated knock-down, showing that MAPP identifies not only the correct sequence motif, but also the binding site position-dependent impact of the RBP on mRNA processing. Applying MAPP to >400 RBP knock-down experiments from the ENCODE project, we have identified multiple pyrimidine motif-binding RBPs that seemingly explain changes in both exon inclusion and poly(A) site choice. Finally, to demonstrate the ability of MAPP to capture meaningful signals from tissues, we have applied MAPP to glioblastoma (also called GBM), a cancer type in which large numbers of pre-mRNA processing changes were reported previously[12,13]. MAPP reveals that the PTBP1 and RBFOX (RNA Binding Fox-1 Homolog) RBPs co-regulate the splicing of hundreds of cassette exons, some of which have already been reported to drive GBM development and progression. In summary, in this study, we have developed MAPP and demonstrated that "MAPPing" RNA-Seq experiments enables the identification of key pre-mRNA regulators, their binding motifs and functions, as well as their role in healthy and diseased cellular states.

## Results

### MAPP infers impact maps for pre-mRNA processing regulators

Whereas RBPs have long been known to orchestrate pre-mRNA splicing (e.g., ref. [14]), their impact on 3′ end processing has only recently started to become apparent[4,5,15,16], giving rise to the question of whether RBP regulators act in a coordinated manner on both splicing and 3′ end processing[6,16]. A bottleneck in addressing this question is that compared to other types of regulators, such as transcription or epigenetic factors, the fraction of RBPs with well-characterized binding specificities is relatively minor. In addition, even for those RBPs for which binding specificities have recently been characterized with high-throughput experiments, the impact and mode of action on pre-mRNA processing remain speculative. To address such questions, we have developed MAPP (Fig. 1).

MAPP makes use of a powerful functional concept that we have previously exploited in our KAPAC tool[6], namely explaining relative expression levels of transcript isoforms across samples with sequence motifs located in nascent transcripts. In contrast to KAPAC, which implemented only the final step of inference of RBP impact on polyadenylation, MAPP provides an end-to-end solution to the inference of motifs, known or not to bind specific RBPs, that impact splicing, 3′ end cleavage or both processes.

MAPP includes a computational model, MAEI (Motif Activity on Exon Inclusion), designed to infer the position-dependent activity of sequence elements on cassette exon inclusion, along the KAPACv2.0 model that infers the activity of motifs on poly(A) site processing which builds upon our previously described KAPAC approach[6]. While similarly to KAPAC MAPP considers the entire space of sequence motifs, modeled as k-mers, that could impact pre-mRNA processing, its functionality is more general, as it can also work with position-dependent weight matrix (PWMs, see Methods) representations of known RBP binding specificities. The two modules model changes in exon inclusion and poly(A) site usage across genes as functions of the motif counts within regions located at various distances from the processed sites. More specifically, given RNA-Seq data from a cellular system of interest (Fig. 1a, b), MAPP first infers the level of inclusion of alternatively spliced exons and the usage of distinct poly(A) sites. For the latter, it makes use of our previously developed PAQR tool[6] (Supplementary Figs. S1, 2, 26). Then, the MAEI and KAPACv2.0 models are fitted to the corresponding pre-mRNA processing event data to identify sequence motifs that can explain global splicing and poly(A) site usage patterns, respectively (Fig. 1c). By applying the models to sequence windows located at specific distances relative to pre-mRNA processing sites (50 nts sliding by 25 for all our analyses unless specified otherwise), position-dependent activity z-scores are inferred for each motif. MAPP ranks the sequence elements based on their significance and reports the position-dependent z-scores in the form of impact maps[6] (Fig. 1d), which provide detailed insights into the activity (activating or repressive) as well as the position dependency of specific RBPs. Importantly, as MAPP can infer impact maps for motifs known to correspond to specific regulators, as well as for motifs that have not yet been linked to a specific RBP, it is able to detect sequence-specific regulatory activities that point to previously unknown regulators.

### Both the binding specificity and the position-specific impact of known regulators are uncovered de novo by MAPP

To validate MAPP, we applied it to datasets from experiments where proteins with a known effect on splicing/polyadenylation were perturbed. We started with the well-characterized HNRNPC RBP and found that the sequence motif most significantly associated with both the measured changes in exon inclusion as well as poly(A) site usage is penta-U, the motif that was previously confirmed by multiple studies to be the primary binding motif of HNRNPC[17–19] (Fig. 2a, top panel). The PWM representing this motif had the largest combined z-score out of the 344 PWMs that we curated from the ATtRACT database (see Methods). Also, the impact map inferred by MAPP is highly consistent with prior reports. That is, in control (CTRL) samples, where the expression of HNRNPC is high, MAPP infers a repressive effect (marked as blue squares) on 3′ splice site (3′SS), 5′ splice site (5′SS) and polyadenylation site (PAS) processing. Conversely, these sites are processed, and thus, the activity of the penta-U motif is positive in the knock-down cells, where the expression of HNRNPC is low. These results are supported by a multitude of studies (e.g., refs. [5,19,20]). To determine whether the differentially processed sites are indeed bound by the suggested RBPs, we further analyzed data from enhanced crosslinking and immunoprecipitation (eCLIP) experiments from the ENCODE project[21,22]. Towards this, we have selected the top 200 3′SS, 5′ SS, and PAS whose usage changes most in the expected direction, upon HNRNPC knock-down, as well as the 1000 sites that change least. For these sites, we have constructed position-dependent coverage profiles for HNRNPC eCLIP data. The resulting profiles indicate that HNRNPC is indeed regulating splicing and polyadenylation via direct interaction with the RNAs at the regions inferred by MAPP (Fig. 2a and Supplementary Fig. S3a).

We next turned to a well-characterized splicing regulator, the RBFOX1 RBP. Analyzing data from an experiment where the RBFOX1-dependency of exons was determined in RBFOX2-deficient HEK293 cells in which RBFOX1 was inducibly expressed from a Flp-in locus[23], MAPP ranks the previously described RBFOX1-binding sequence, UGCAUG, as the most significant in explaining exon inclusion, further inferring that it has an activating activity when located downstream of 5′SS[24] (Fig. 2b and Supplementary Fig. S3b). MAPP also highlights the opposite activity near the 3′SS, where RBFOX1-binding motifs are

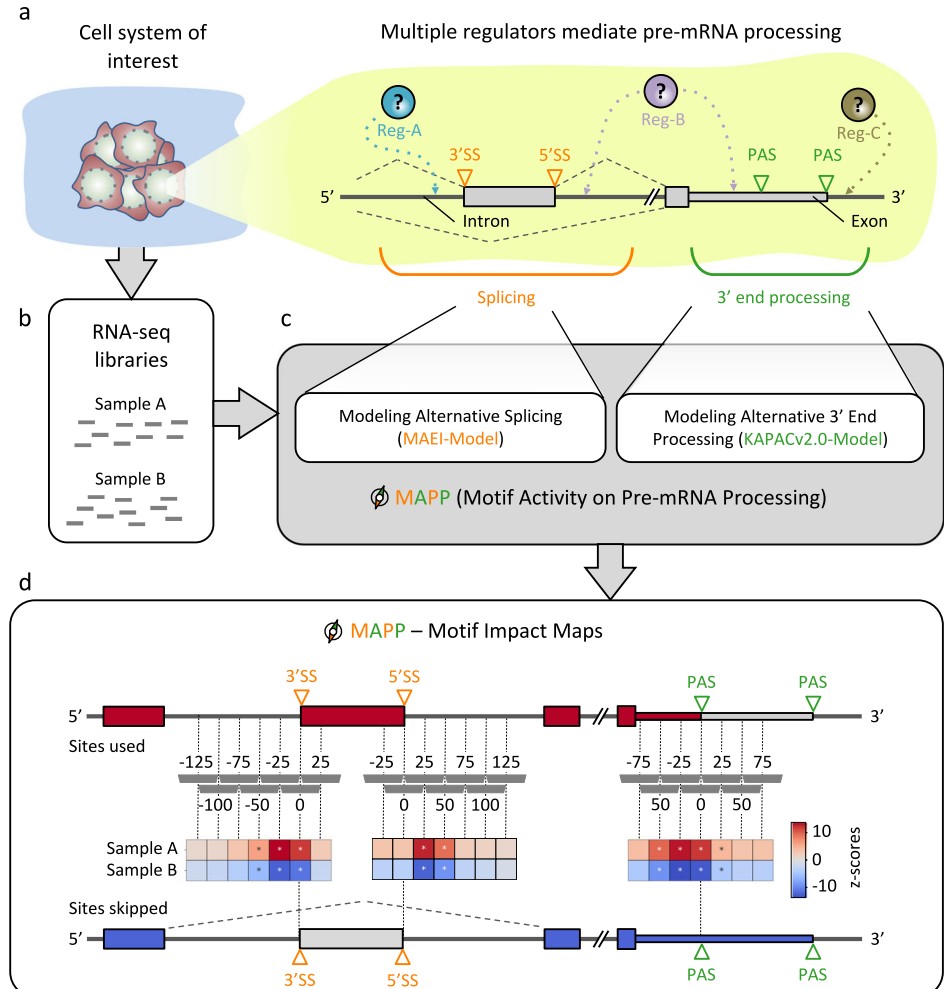

**Fig. 1 | Inferring maps of RBP impact on splicing and 3′ end processing with MAPP. a** Sketch illustrating how regulators (Reg) bind pre-mRNAs to influence the usage of splice sites (SS) and/or poly(A) sites (PAS). **b** RNA sequencing (RNA-seq) libraries are available or can be created for most cellular systems of interest. **c** MAPP analyzes the splicing and 3′ end processing patterns apparent in the RNA-Seq data with the MAEI (Motif Activity on Exon Inclusion) and KAPACv2.0 (K-mer Activity on PolyAdenylation site Choice version 2.0) models, respectively. **d** MAPP infers regulatory motifs for RBPs and reports detailed maps of their position-dependent impact on cassette exon inclusion and poly(A) site usage, respectively, by applying the models to genomic windows located at specific distances relative to the RNA processing sites (dashed gray vertical bars).

associated with reduced exon inclusion. While this repressive effect appears to be much weaker compared to the activating effect of motifs located downstream of 5′SS, it is interesting to infer simply from the RNA-Seq data that RBFOX1, like other RBPs[25], can have opposing impacts depending on the location of binding sites. These results demonstrate that by making use of standard RNA-Seq experiments only, MAPP enables fine-grained insights into the binding-specificity and position-dependent impact of RBPs on splicing and 3′ end processing.

**MAPP impact maps unveil the regulation code of multiple RBPs**
Next, we used the large array of RBP knock-down datasets available from the ENCODE project to comprehensively infer the sequence specificity, binding site position-dependent impact, and activating or repressive mode of action of human RBPs on pre-mRNA processing. Applying MAPP to 456 RBP knock-down experiments available in ENCODE, we found that the tool is also here able to identify the motif known from the ATtRACT database to correspond to the protein whose expression was altered in the experiment. Figure 3a shows summary results for samples for which the ATtRACT-provided PWM for the targeted RBP was ranked among the top five most significant motifs. As the ATtRACT-provided PWMs corresponding to the

perturbed proteins were not always the most significant motifs in explaining the RNA processing alterations, we also ran MAPP in the k-mer mode, to determine which sequence elements explain best the observed changes. For some RBPs, such as PCBP1 and HNRNPK, the k-mer-based results are more significant and consistent compared to the PWM-based results, indicating that the inferred k-mer better represents the RBPs binding specificity than the PWM available in public databases. Interestingly, MAPP uncovers that the general splicing factor SRSF1 and the PCBP1 RBP promote splice site processing, while other RBPs (e.g., HNRNPC, PTBP1, and HNRNPK) appear to have a repressive role. Half of the RBPs (HNRNPC, PTBP1, PCBP1, and HNRNPK) regulate both splicing and polyadenylation by acting on similar sequence motifs. Moreover, the RBPs generally have the same type of activity—activating or repressive—on both splicing and poly-adenylation. Thus, RBPs inferred by MAPP to have a dual role, in splicing and polyadenylation, appear to predominantly act as either activators or repressors in both pre-mRNA processing steps. This may also hint at a concerted regulation of alternative terminal exons by individual regulators, but it must go beyond the regulation of terminal exons, because, in many cases, the motifs have similar activity around the 5′SS, which does not occur in terminal exons. To investigate whether 3′ end processing factors might also impact cassette exon splicing

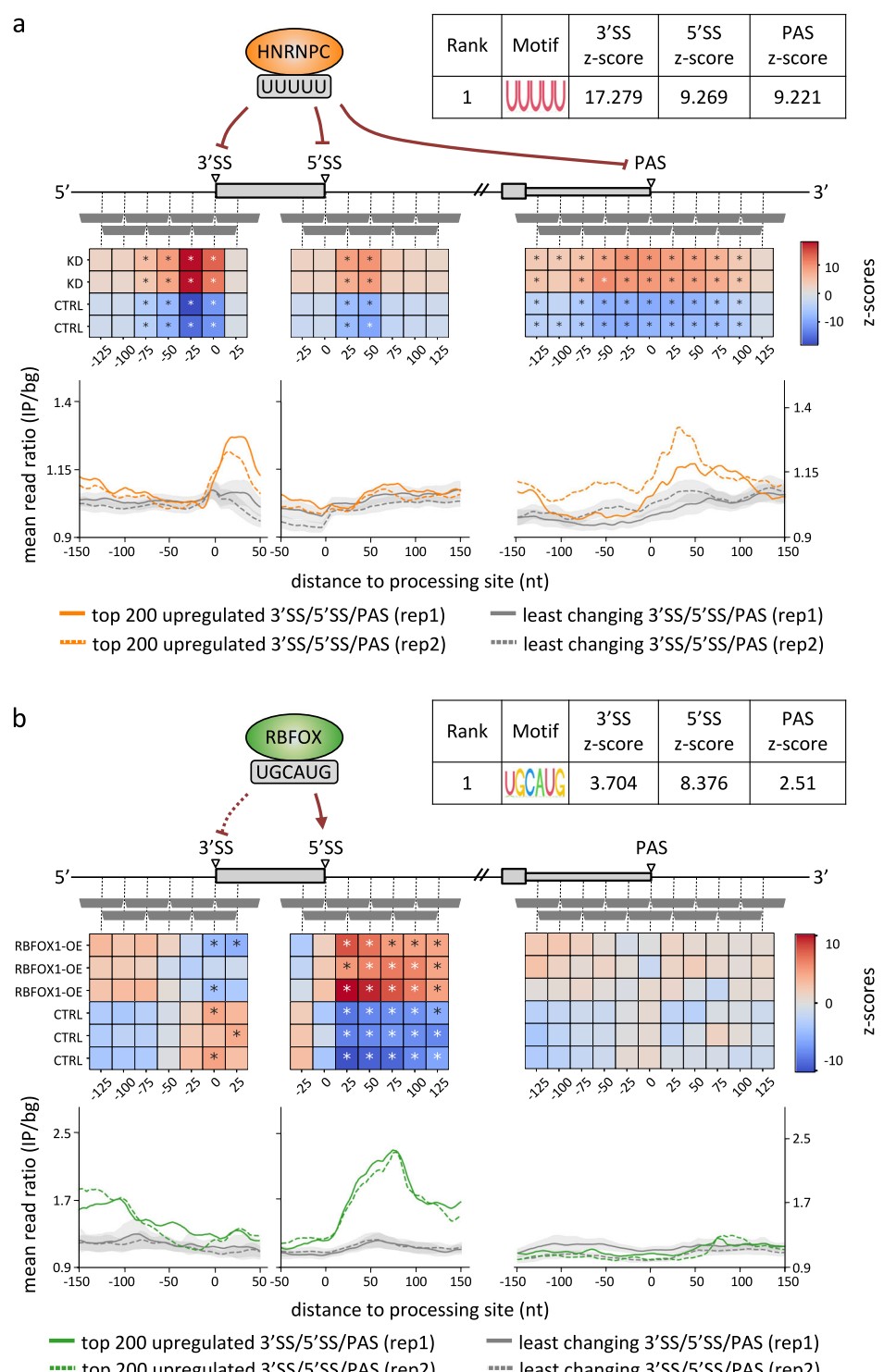

**Fig. 2 | MAPP infers the known regulatory impact of the HNRNPC and RBFOX RBPs on splicing and 3′ end processing from RBP expression perturbation datasets. a** Top panel: z-scores of activity changes in the vicinity of 3′ splice sites (3′ SS), 5′ splice sites (5′SS), and poly(A) sites (PAS) inferred from an HNRNPC knock-down dataset[18]. The PWM with the highest inferred combined z-score of all PWMs has the penta-U motif as consensus. By fitting the splicing and 3′ end processing models of MAPP to overlapping windows (horizontal gray bars) located at specific distances relative to splice and poly(A) sites, position-dependent activity z-scores are inferred. Statistically significant z-scores are marked with an asterisk. Bottom panel: Smoothened (±5 nt) HNRNPC eCLIP-based coverage profiles in the vicinity of the top 200 3′SS, 5′SS, and PAS, whose usage is most upregulated (orange) or does change least (gray) upon HNRNPC knock-down. Least-changing targets are pre-sented as an average of mean per-position coverage calculated over non-targets sampled 100 times with repetition ±1 standard deviation. **b** Top panel: MAPP results as described in a, but here applied to a RBFOX2-deficient HEK293 cell line with induced expression of RBFOX1 which is known to regulate splicing at 5′SS by binding to UGCAUG sequences[24]. Bottom panel: eCLIP profiles as in a, but for the RBFOX2 RBP in the vicinity of the top 200 3′SS, 5′SS, and PAS, whose usage is most upregulated (green) or does change least (gray) upon RBFOX1 overexpression.

a

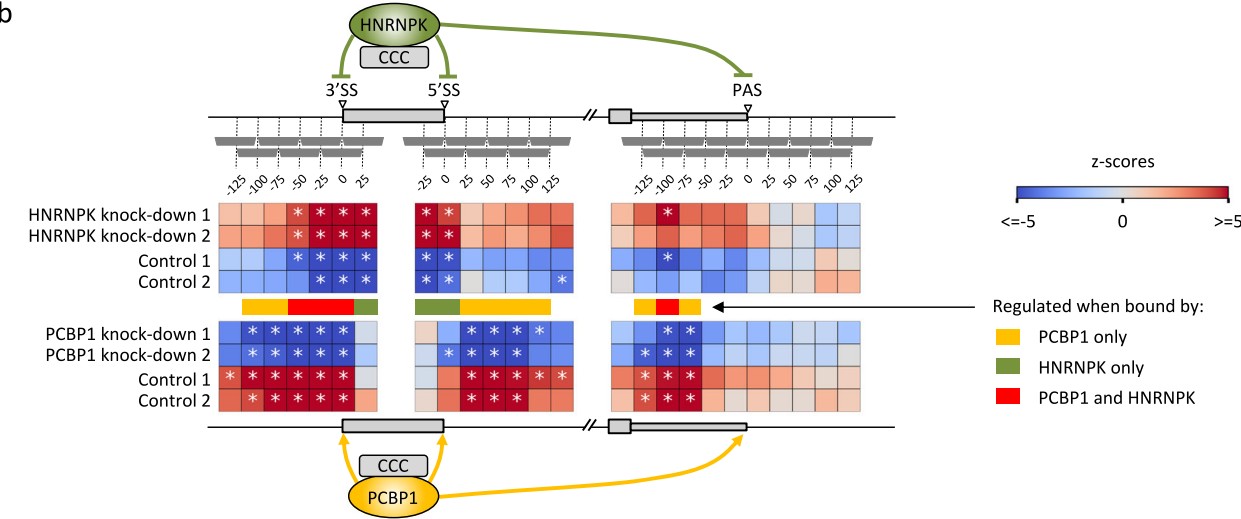

b

**Fig. 3 | MAPP reveals the concerted impact of pre-mRNA processing regulators on splicing and polyadenylation. a** For each RBP, we determined the first motif (in the order of MAPP-provided significance) that is assigned in the ATtRACTdb to the RBP that was depleted in each experiment. Column 3 shows the rank of that motif as inferred by MAPP. The table contains RBPs where the known binding motif was among the top five reported by MAPP. The activity profiles are shown similarly to those in Fig. 2, with the top two rows indicating the knock-down and the bottom two the control conditions. Windows within regions around 3′SS, 5′SS, and PAS are set to the same ranges as done in Fig. 2. The central window sliding through a given RNA processing site (−25nt,+25nt) was marked as a black square in the legend (bottom right). Furthermore, in addition to the PWM-based MAPP runs, we have carried out a similar analysis exploring all possible k-mers of length 3 to 5. The top-ranked k-mer is reported for each experiment alongside the corresponding PWM result. **b** MAPP impact maps of selected HNRNPK (ENCSR853ZJS) and PCBP1 (ENCSR635FRH) RBP knock-down and control experiments. Colored bars indicate regions in which exclusively HNRNPK (green), exclusively PCBP1 (yellow), or both RBPs (red) regulate pre-mRNA processing by binding the "CCC" k-mer.

we further applied MAPP to CFIm68 and CFIm25 perturbation experiments. As expected, MAPP infers that the well-known CFIm binding sequence UGUA has a strong activating effect on 3′ end processing, but we observed no significant impact on alternative splicing (Supplementary Fig. S4). Finally, MAPP also infers that RBPs with similar sequence specificity can exert their regulatory roles by binding to the pre-mRNA in different positions relative to the pre-mRNA processing sites (Supplementary Fig. S5). For instance, both PCBP1 and HNRNPK bind a 'CCC' sequence element to regulate splicing and polyadenylation. However, while the repressive impact of HNRNPK on cassette exon splicing seems to be focused on the immediate vicinity of the processing sites and exonic regions, PCBP1 appears to activate splicing from a broader intronic region (Fig. 3b).

While the proteins with known PWMs shown in Fig. 3a have been implicated in splicing before, we also investigated cases where MAPP identified a significant k-mer able to explain the pre-mRNA processing changes, but not one of the considered PWMs. One interesting example is the Poly(U) Binding Splicing Factor 60 (PUF60) RBP. ENCODE provides knock-down experiments for this protein in two cell lines: K5643 (ENCSR558XNA) and HepG2 (ENCSR648BSC). The two experiments that yield the most significant MAPP results consistently infer highly similar U-rich sequence elements (Supplementary Fig. S6), corresponding to the motif inferred to be bound by PUF60 in vitro, in RNA Bind'n-seq experiments[21,22]. As PUF60 exhibited a narrow window of activity upstream of 3′ splice sites, we have rerun MAPP at a higher resolution, using windows of 20 nts sliding by only 10 nts. The resulting MAPP impact maps show that the PUF60 RBP is only active when binding to U-rich regions located within a very narrow window (30 to 10 nt) upstream of the 3′ splice site (Supplementary Fig. S6). While this is consistent with a previous report of PUF60 activating exon inclusion by binding to U-rich regions upstream of 3′ splice sites[26], MAPP reveals the very narrow window of PUF60 activity, the intronic region of ~30–10 nts upstream of the 3′ splice site, thus being much more position-specific compared to other regulators mentioned above (Fig. 3). These results illustrate the utility of MAPP in elucidating the position- and sequence-dependent regulation of pre-mRNA processing by RBPs.

## MAPP infers RBPs that drive the oncogenic splicing program active in glioblastomas

As key factors in the post-transcriptional regulation of gene expression, RBPs have been reported to play an important role in numerous diseases, including cancer[27]. In a previous study we have uncovered that the PTBP1 RBP best explains the global remodeling of 3′ UTR length in glioblastoma[6]. Importantly, PTBP1 was previously mainly studied in the context of splicing, and the results from our ENCODE screening recapitulate these results (Fig. 3a). To follow this further, we applied MAPP to a high-quality PTBP1 knock-down dataset without PTBP2 background[28] confirming that PTBP1 does act as global splicing and 3′ end processing regulator (Fig. 4a, bottom panel). Specifically, in addition to its repressive activity on poly(A) site usage, PTBP1 represses the processing of 3′SS and, to some extent, 5′SS. Moreover, from the MAPP impact maps, we can conclude that PTBP1-binding motifs located within the cassette exon itself or the first ~75 intronic nt upstream of the 3′SS are associated with reduced exon inclusion when the expression levels of PTBP1 are high, i.e., in control conditions. These inferences are also supported by PTBP1 eCLIP data (Fig. 4a and Supplementary Fig. S7).

To uncover which regulators can best explain splicing in glioblastomas, we next applied MAPP to cancer samples[29], where it inferred that the PTBP-binding motif has the most significant activity on pre-mRNA processing (Supplementary Data 4, 5) with a motif ranking and position-dependent activity that matches the profile obtained from the PTBP1/2 knock-down data (Fig. 4a, b and Supplementary Data 6). The additional activity of PTBP1 on splicing further

strengthens the case for PTBP1 as a main regulator of pre-mRNA processing in glioblastomas, where PTBP1 is highly expressed[30]. Also, as expected for a regulator that acts as a repressor of pre-mRNA processing, the expression of PTBP1 anti-correlates with its motif activity (Fig. 4c). Interestingly, besides the PTBP1 motif, the RBFOX-associated motif was also identified by MAPP as being differentially active in GBM compared to normal brain samples (Fig. 4b). Moreover, a k-mer-based MAPP run confirmed that in addition to PTBP1-associated CU-rich k-mers, the GCAUG sequence bound by the RBFOX RBPs is also among the significant 5′SS-proximal k-mers that regulate exon inclusion (Supplementary Data 4). Consistently with the known role of RBFOX RBPs as activators of splice site usage, the MAPP-inferred activity correlates remarkably well with RBFOX expression, which is low in GBM and high in healthy brain (Fig. 4c). Applying MAPP to GBM samples from two additional RNA-Seq cohorts[31,32] recapitulated both the PTBP1 and RBFOX expression patterns and the CU-repeat and GCAUG k-mer activities (Supplementary Figs. S8, S9), thereby confirming the PTBP1 and RBFOX RBPs as global regulators of the oncogenic splicing program acting in glioblastomas[30,33].

## Multiple oncogenic splicing events take place downstream of the PTBP1 and the RBFOX RBPs in glioblastomas

Investigating the inclusion level (measured as percent-spliced-in (PSI) across transcripts covering the respective genomic regions) of exons having predicted binding sites for the PTBP1, the RBFOX RBPs, or for both RBPs within the MAPP-inferred regions, we found that cassette exons that are coregulated by both RBPs exhibit the largest differences in PSI when comparing glioblastoma to normal brain tissue (Fig. 5a). Importantly, the average change in exon inclusion increased with the motif-binding score for PTBP1 and RBFOX (Supplementary Fig. S10). The genes with differentially included cassette exons having PTBP1 and RBFOX binding sites in the MAPP-inferred impact regions are highly enriched in the synaptic signaling Gene Ontology (GO) category (Fig. 5b). This suggests that cassette exons that are spliced-in as a result of low PTBP1 and high RBFOX splicing activities in normal brain tissue are largely involved in neuron-specific functions. Importantly, both RBPs were previously reported to regulate brain-specific micro-exon inclusion in healthy brain tissue[34], suggesting that the dysregulation of these two factors in glioblastomas leads to a pattern of exon inclusion that corresponds to less differentiated and specialized cells.

Interestingly, there are multiple splicing events previously reported to be oncogenic among the cassette exons that are differentially spliced-in in normal versus malignant brain tissue and contain binding sites for the PTBP1 RBP, the RBFOX RBPs or both (Fig. 5c and Supplementary Data 7), thereby providing connections between the dysregulation of the PTBP1 and RBFOX RBPs in GBM and the downstream effects on malignant cellular behavior.

Exon 6 of the *ANXA7* gene (Supplementary Data 7), skipped in GBM (Fig. 5c), provides an interesting positive control for our analysis. Inclusion of this exon was previously shown to be regulated by PTBP1, and skipping of *ANXA7* exon 6 promotes the progression of GBM by fostering angiogenesis[35]. Thus it provides an interesting link between the MAPP-inferred activities and the molecular properties of GBM. *ANXA7* exon 6 has binding sites for the PTBP1 RBP in the regions that MAPP infers to be regulated by the PTBP1 RBP (Supplementary Data 7). Another interesting example is exon 16 of the *NF2* gene, whose inclusion level is reduced in GBM (Fig. 5c). This exon has binding sites for RBFOX, but not PTBP1 (Supplementary Data 7). Previous studies have shown that there exist two major *NF2* isoforms, isoform 1, which does not contain exon 16, and isoform 2, which does. Even though the exact role of these isoforms is still a matter of debate, both of them have been implicated in cancer development[36,37].

Besides these two examples of differentially included cassette exons, having binding sites for the PTBP1 or the RBFOX RBPs, respectively, there exist also hundreds of exons that are skipped in

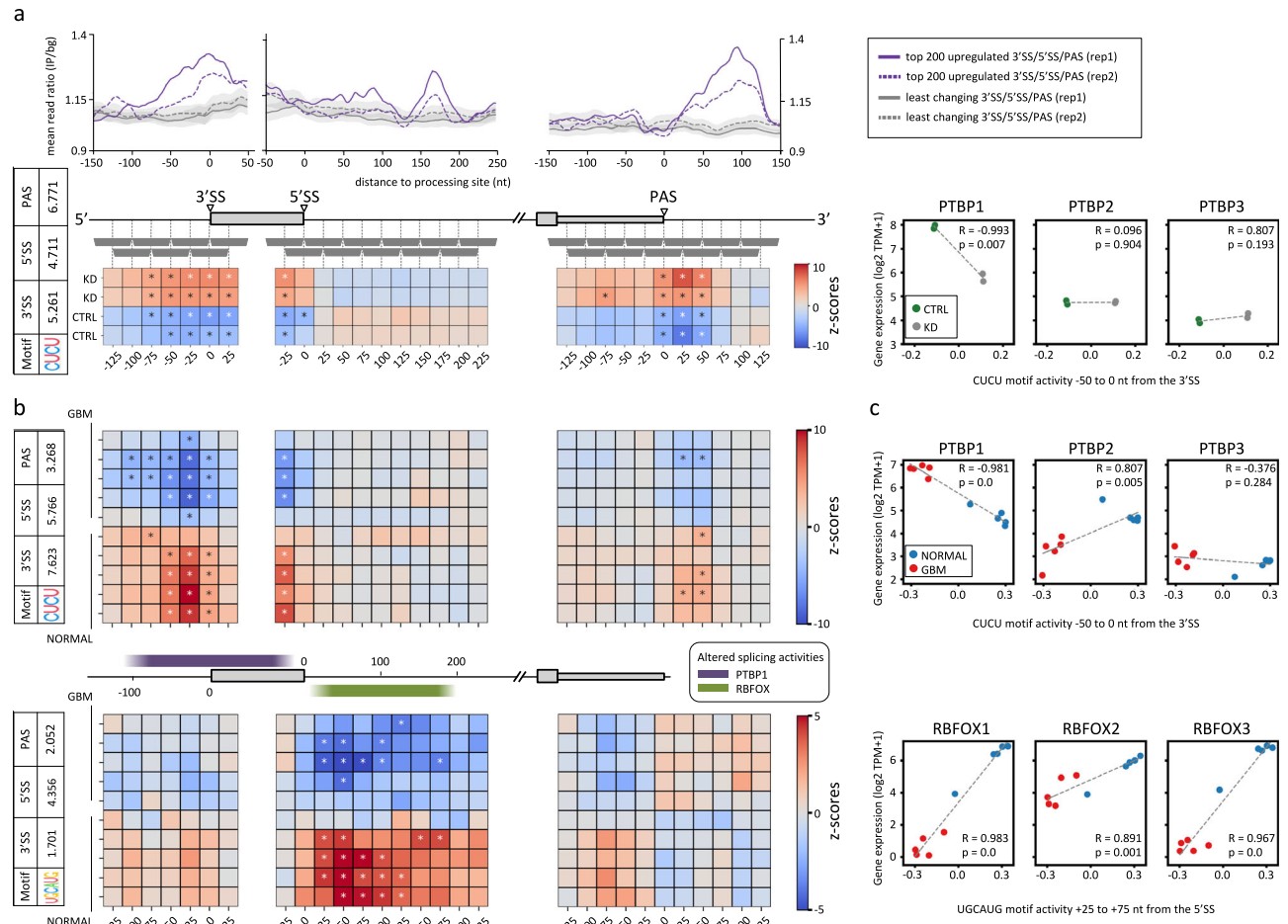

**Fig. 4 | MAPP unravels the joint effect of the PTBP1 and RBFOX RBPs on pre-mRNA processing in glioblastoma. a** PTBP1 eCLIP densities around pre-mRNA processing sites calculated and presented as in Fig. 2 (top panel) and impact maps for the PTBP-bound CUCU motif as inferred by MAPP (bottom left panel) from control cells (CTRL; $n = 2$) and cells depleted of both PTBP1 and PTBP2 by siRNA-mediated knock-down (KD; $n = 2$). Bottom right panel: PTBP1/2/3 expression versus the activity of the CUCU motif within 50 nt upstream of 3′ splice sites (3′SS). R indicates the Pearson correlation coefficient and $p$ the corresponding two-tailed $p$ value. The dashed gray line shows the linear regression. **b** MAPP results for

glioblastoma (GBM; $n = 5$) and normal brain (NORMAL; $n = 5$) samples for the PTBP-bound CUCU motif (top panel) as well as for the RBFOX-bound UGCAUG motif (bottom panel). Regions with statistically significant CUCU motif activity (purple) or UGCAUG motif activity (green), respectively, are highlighted in the cartoon (mid panel). MAPP was run without a minimum exon length constraint in order to also account for micro-exons prevalent in neurons. **c** Scatter plots as outlined in (**a**), but for the samples described in (**b**) showing the mRNA expression levels of the PTBP and RBFOX RBPs, respectively, versus the MAPP-inferred activities for the RBP-corresponding motifs within the indicated region windows.

GBM and that have binding sites for both RBPs within the MAPP-inferred regions (Fig. 5a and Supplementary Fig. S10–12). Of these, one interesting candidate is exon 3 of the *RTN4* gene (Supplementary Data 7). Consistent with our observations in GBM (Fig. 5c), a previous study has shown that high levels of PTBP1 cause *RTN4* exon 3 skipping. In contrast, PTBP1 knock-down results in enhanced inclusion of *RTN4* exon 3. Importantly, the overexpression of the exon 3-including RTN4 splice isoform was shown to decrease the proliferation of glioma cell lines, whereas skipping of *RTN4* exon 3, as observed in GBM (Fig. 5c), contributes significantly to their rapid growth characteristics[38].

Importantly, in contrast to the high PTBP1 and low RBFOX RBP levels that we observe in malignant brain tissue (Fig. 4c and Supplementary Figs. S8b, 9b), low levels of PTBP1 and high levels of RBFOX contribute to the splicing program of healthy neurons[34]. Consistently, MAPP infers PTBP1 and RBFOX both having a high motif activity in brain tissue relative to most other tissues (Supplementary Fig. S13). We also found a reversal of PBX Homeobox 1 (PBX1) exon 7 inclusion in GBM compared to healthy neurons[39] (Fig. 5c). This exon too contains binding sites for both RBPs within the regions inferred by MAPP

(Supplementary Data 7). Interestingly, in mouse embryonic stem cells (ESCs) induced expression of the PBX1 isoform containing exon 7 activates the transcription of neuronal genes[39].

Another interesting exon having binding sites in the MAPP-inferred regions for both RBPs is exon 10 of *PTBP2*. It is known that PTBP1, abundantly expressed in undifferentiated neural stem cells (NSCs), is downregulated during neuronal differentiation, while its PTBP2 paralog is upregulated, leading to the increased inclusion of neuron-specific exons[40,41]. High expression of PTBP1 in NSCs and undifferentiated precursors promotes skipping of the *PTBP2* exon 10 and indeed, this is what we observe in the GBM samples relative to normal brain tissues (Fig. 5c). Interestingly, skipping of *PTBP2* exon 10 has been linked to the binding of RBFOX and results in transcript isoforms with a premature stop codon that are subject to degradation by nonsense-mediated mRNA decay[33]. Consistently, we observe reduced *PTBP2* mRNA expression in GBM relative to the normal brain tissue samples (Fig. 4c and Supplementary Figs. S8b, 9b).

Thus, the high expression of PTBP1 in glioblastomas relative to normal brain tissue might contribute to a less differentiated state, which was suggested to be the origin of glioblastoma[42], and is also one

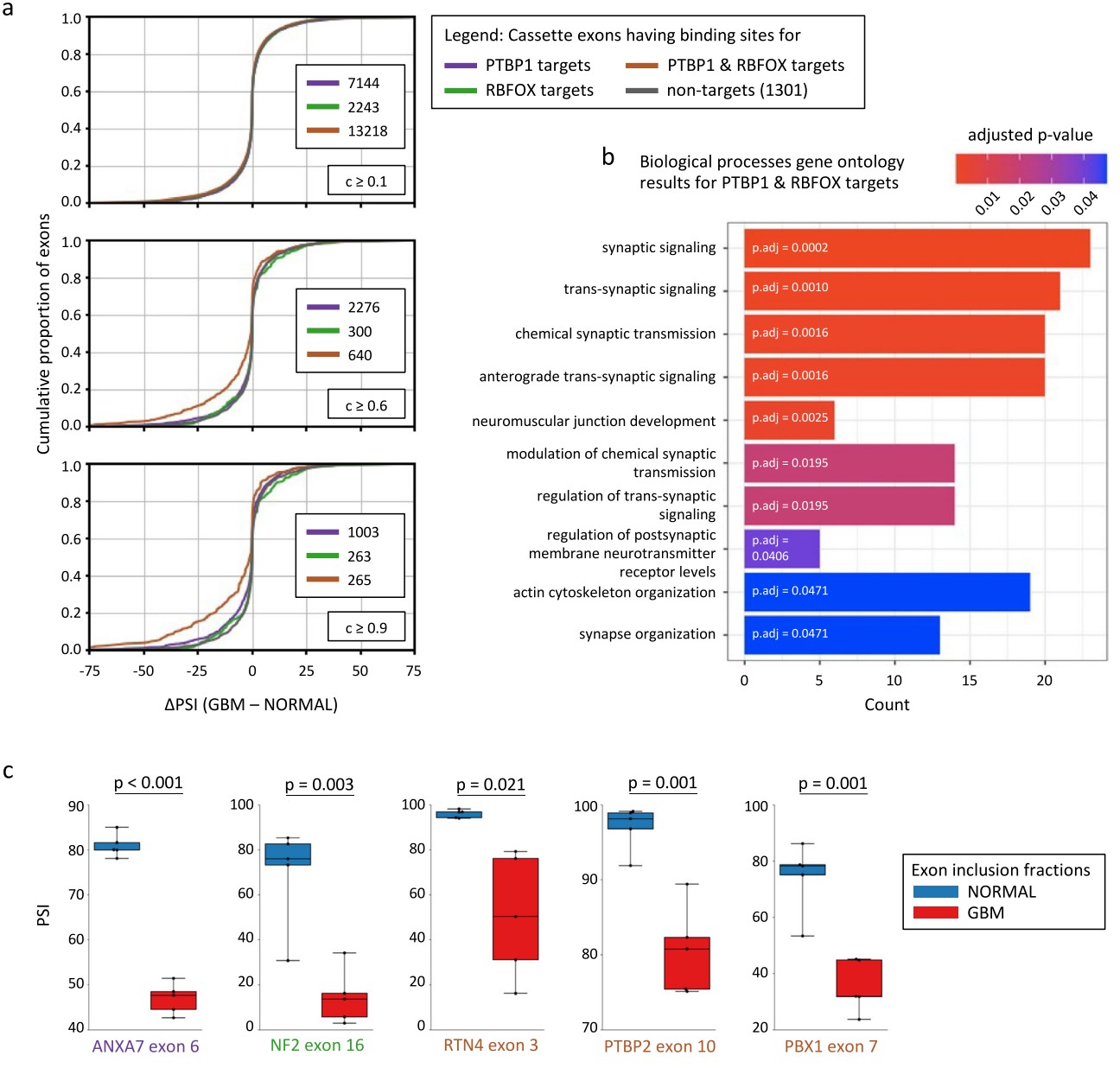

**Fig. 5 | Concerted effects of the PTBP1 and RBFOX RBPs regulate cassette exon skipping in glioblastomas. a** Distribution of differences between percent-spliced-in averages (ΔPSI) observed in glioblastoma (GBM; n = 5) versus normal brain (NORMAL; n = 5) for cassette exons regulated by PTBP1 and/or RBFOX according to their binding probabilities within the regions inferred by MAPP to be significantly regulated, using increasing binding probability (c ≥) cutoffs. The number of cases in each group is indicated. **b** Top ten enriched gene ontology (GO) terms in the category Biological Processes as inferred for genes with cassette exons that have binding sites for both RBPs (with binding probability ≥0.6) and that are differentially included in glioblastoma compared to normal brain samples. The list of differentially included exons was obtained by running a two-sided Welch t-test.

*P* values for both differentially included exons and enriched GO terms were adjusted for multiple comparisons using the Benjamini-Hochberg method. A detailed description of the GO analysis is provided in Supplementary Information (2.12). **c** Percent-spliced-in (PSI) for five cassette exons that have been associated with cancer-related isoforms previously. *p* indicates the *p* value of comparing normal (NORMAL; n = 5) and glioblastoma (GBM; n = 5) samples using a two-tailed *t*-test, not assuming equal variances. The text color represents RBP regulation as defined in (**a**). Boxes indicate the interquartile range (IQR), with the black line corresponding to the median and whiskers extending to the most extreme values. Individual data points are shown as black dots.

of the general hallmarks of cancer[43]. The cell type at the origin of GBM that acquires the initial oncogenic mutations is still unknown, although recent research suggests that this cell type might be NSCs, which reside in the ventricles of the brain during development[44]. To investigate this further from a splicing perspective, we used MAPP, comparing cells from the ventricular zone (VZ) to cells from the cortical plate (CP)[45]. Interestingly, MAPP revealed similar splicing and expression patterns as observed in GBM versus healthy brain tissue. This suggests that the PTBP1 and RBFOX-dependent splicing program

active in glioblastomas relative to healthy brain tissues is indeed similar to the one active in cells from the VZ relative to the further differentiated cells residing in the CP (Fig. 6a–c). Further we applied MAPP to a human neural stem cell model (hTERT immortalized NSCs) carrying only one of the alterations thought to contribute to GBM (IDH1 R132H)[46]. As expected, the NSC model showed splicing patterns that are comparable to GBM and undifferentiated cells residing in the VZ (Fig. 6d–f), further hinting towards GBM cells representing a less differentiated cellular state in terms of their global splicing pattern,

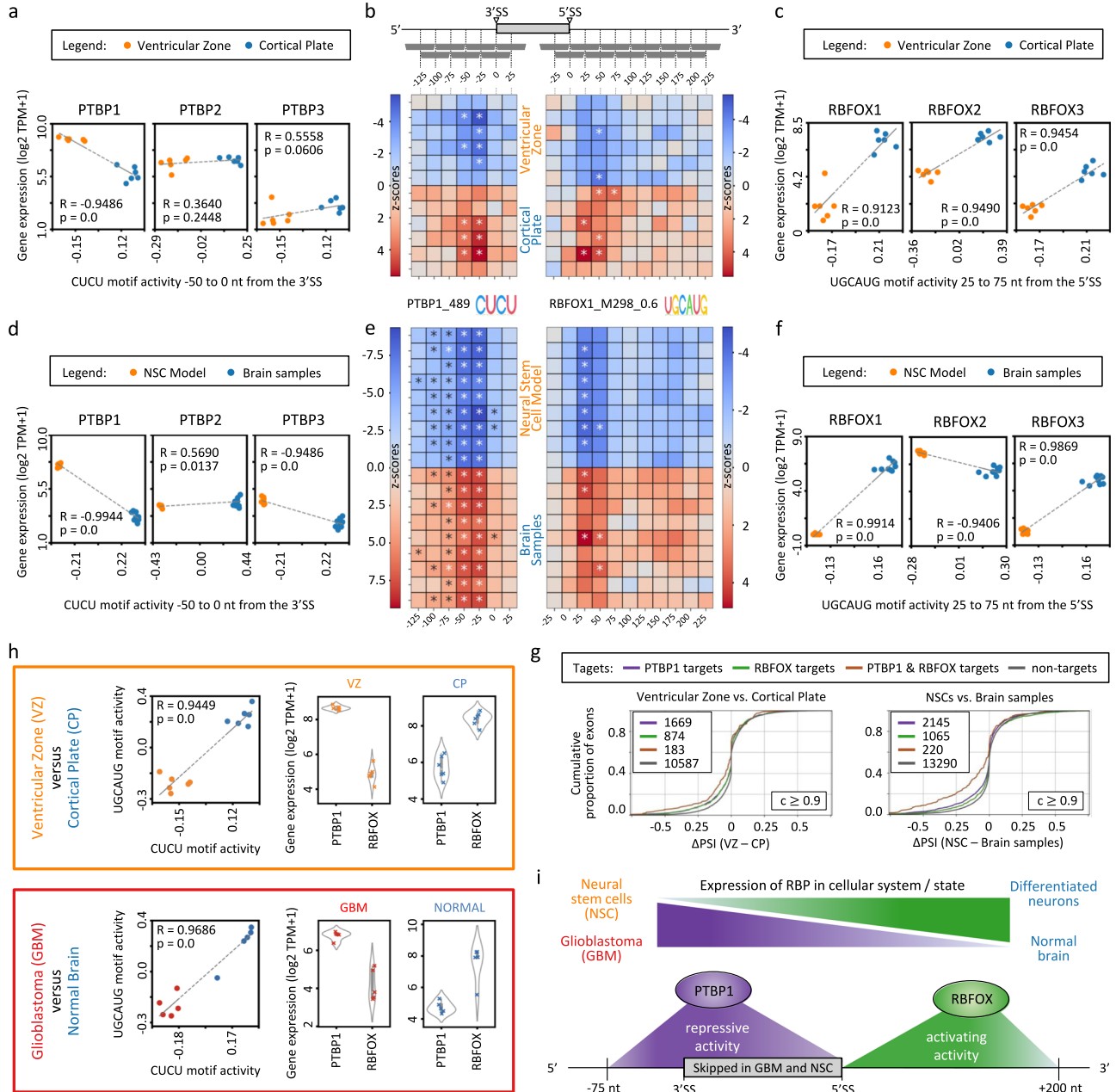

**Fig. 6 | Regulation of cassette exons in neuronal differentiation and glioblastomas. a** Scatter plots of the PTBP RBPs mRNA expression levels versus the activities of the PTBP-associated CUCU motif for the indicated region window as inferred by MAPP for cells from the ventricular zone ($n = 6$) and cortical plate ($n = 6$). R indicates the Pearson correlation coefficient and $p$ the corresponding two-tailed $p$ value. The dashed gray line shows the linear regression. **b** MAPP-inferred impact maps of the PTBP-associated motif at 3′ splice sites (3′SS) and the RBFOX-associated UGCAUG motif at 5′ splice sites (5′SS) for the two cell populations described in (**a**). **c** Scatter plots as in (**a**) but for the RBFOX RBPs mRNA expression levels versus the activities of the RBFOX-associated motif for the indicated region window as inferred by MAPP for the cell populations described in (**a**). **d** Scatter plots as in (**a**) but for a neural stem cell model ($n = 9$) and normal brain samples ($n = 9$). **e** MAPP-inferred impact maps as in (**b**) but for the cells described in (**d**). **f** Scatter plots as in (**c**) but for the cells described in (**d**). **g** Distribution of differences

between percent-spliced-in averages (ΔPSI) observed in the cells described in (**a**) (left panel) and for the cells described in (**d**) (right panel) for cassette exons regulated by PTBP1 and/or RBFOX according to having a binding probability of ≥0.9 within the regions inferred by MAPP to be significantly regulated. The number of cases in each group is indicated. **h** Comparison of motif activities (in the windows indicated in **a**, **c**, **d**, **f**) and expression levels of PTBP1 and the RBFOX RBPs (sum of RBFOX1/2/3) in cells residing in the ventricular zone (VZ; $n = 6$) and the cortical plate (CP; $n = 6$) (top panel) to those observed in glioblastoma (GBM; $n = 5$) and normal brain (NORMAL; $n = 5$) (bottom panel). Wide boxes in the violin plots indicate the interquartile range (IQR), with the white dot inside corresponding to the median and whiskers indicating 1.5 times the IQR from the hinge. Data points are shown as colored dots. **i** Graphical summary of cassette exon coregulation by the PTBP1 and RBFOX RBPs in the indicated cell systems.

which appears to be broadly driven by the PTBP1 and RBFOX RBPs (Figs. 5a and 6g). In summary, the similarities of the MAPP-inferred splicing regulation (Fig. 6h) further strengthens the current view that GBM represents cells in a more stem-like state compared to healthy neurons[43,47] (Fig. 6i).

## Discussion

By binding to sequence elements in (pre)mRNAs, RBPs regulate gene expression at co-transcriptional and post-transcriptional levels. In particular, they can affect both splicing and 3′ end processing, key steps in mRNA maturation. Additionally, the interaction of RBPs with

mature mRNAs can regulate their transport, localization, and translation[48]. Understanding the global and concerted effect of various RBPs on the cellular transcriptome is undoubtedly key to uncovering mechanisms of gene expression dysregulation in various pathological conditions, including cancer[49,50]. In this study, we presented a computational approach for inferring the regulatory impact of various RBPs on splicing and 3′ end processing.

We have validated our method on data pertaining to proteins with well-established roles in splicing and/or polyadenylation. Specifically, the MAPP-inferred activities of the HNRNPC RBP are in line with its previously reported role in preventing the exonization of cryptic *Alu* elements[20,51]. Many *Alu* elements have evolved to become cassette exons, and the potentially deleterious inclusion of these exons into mature mRNAs needs to be tightly regulated. The impact maps constructed by MAPP are fully consistent with this role of HNRNPC (Fig. 2a). MAPP also recovers the previously-noted position-specific regulation of exon inclusion by RBFOX (Fig. 2b), whereby binding of RBFOX upstream of cassette exons results in their exclusion, while binding downstream of such exons promotes their inclusion[52]. While consistent with this model, our results provide higher resolution for the binding site position-dependent effects of RBFOX. Specifically, they indicate a larger impact on the downstream, inclusion-promoting sites. Furthermore, MAPP indicates that binding sites that are located further upstream in the introns also have an overall inclusion-promoting effect, consistent with an earlier report[24]. Thus, MAPP provides direct and broad insight into the activity of RBP binding sites from RNA-Seq datasets, without a need for stratifying the data or determining the binding sites with methods such as crosslinking and immunoprecipitation. MAPP's position-dependent impact maps thus enable an efficient and improved understanding of how RBPs impact their targets.

After benchmarking MAPP on RBPs with known impact on pre-mRNA processing, we turned to the >400 RBP knock-down datasets available from the ENCODE project and revealed that multiple regulators affect exon inclusion and 3′ end processing (Fig. 3). Once again, thanks to the sliding window approach of MAPP we found that distinct regulators differ not only in their role (which for all investigated RBPs seem to be the same in the two processes, i.e., to either enhance or repress) but also in their position specificity. For example, MAPP not only highlighted the opposite effect of two proteins, HNRNPK and PCBP1, which bind the same "CCC" sequence, on cassette exon inclusion, but also that the distance range of their impact differs: PCBP1 acts more broadly in the introns flanking the cassette exon, while HNRNPK acts in a more focused manner, at the exon-intron boundaries. This could reflect structural constraints on RBP-RNA interactions, given the step of RNA processing where an individual RBP acts and the complexes that it may be part of. Interestingly, many of the investigated RBPs act as repressors of pre-mRNA processing. Given that the sequence elements that are involved in the processing are typically short, our results could indicate that many repressors are needed to mask the many decoy processing sites across the genome[53].

While the knock-down experiments are very informative in revealing the specificity and mechanism of an RBP, the pattern of regulation within the context of a tissue, where multiple RBPs likely vary in concentration in a concerted manner, are more challenging to interpret[54]. Nevertheless, applying MAPP to normal brain and glioblastoma samples, we uncovered many exons that appear to be coregulated by the PTBP1 and RBFOX RBPs (Fig. 5 and Supplementary Figs. S11, S12), two regulators that were reported previously to act in concert[22,34]. MAPP analysis of glioblastoma samples yields impact maps that strikingly resemble those obtained from RBP perturbation experiments for the individual RBPs (Figs. 2b, 4a, b). In glioblastomas, the splicing-activating RBFOX RBPs are downregulated, whereas the PTBP1 splicing repressor is highly expressed compared to normal brain tissue (Fig. 4c). Interestingly, compared to GBM samples, the

analyzed oligodendroglioma samples showed similar but much weaker splicing patterns for the PTBP1 and RBFOX RBPs, whereas the investigated astrocytoma showed patterns that are comparable to normal brain (Supplementary Fig. S9). The usage of many cassette exons alternatively spliced in glioblastomas are repressed at their 5′SS by the highly abundant PTBP1 RBP, whereas the usage of their 3′SS lacks splicing due to the lack of RBFOX RBPs (Fig. 4b and Supplementary Figs. S8, S9). Thus, the oncogenic splicing program of glioblastomas is a result of both overexpression of PTBP1 and the downregulation of RBFOX RBPs compared to healthy brain tissues (Fig. 6i), in which the splicing activities of these RBPs are correspondingly high and low relative to many other tissues[55] (Supplementary Fig. S13). Notably, multiple of the cassette exons predicted to depend on PTBP1 and RBFOX for their splicing in GBM were previously shown experimentally to drive cells into a more malignant state (Fig. 5c). Examples are *PBX1* exon 7 and *PTBP2* exon 10, whose skipping was reported to contribute to less differentiated cellular states[39–41,43]. Further, skipping of the *RTN4* exon 3 was demonstrated to increase cell proliferation of glioma cells[38], and reduced inclusion of exon 6 of the *ANXA7* gene was reported to promote glioblastoma progression[35]. Besides the already experimentally validated oncogenic splicing events, among the large number of cassette exon skipping events that we detected in glioblastomas (Fig. 5a and Supplementary Figs. S10–12), there are most probably further candidates that remain to be characterized towards their involvement in brain tumor development and progression. Importantly, the identification of RBPs that broadly impact mRNA processing in specific conditions, such as cancers, can provide potential entry points for the development of therapies. For instance, a recent study has demonstrated that knocking down PTBP1 in glioblastoma cells promotes their neural differentiation into a non-proliferating cellular state[56]. Thus, downregulation of the PTBP1 RBP provides a promising approach for glioblastoma treatment in the future, which might be further developed and optimized, e.g. by parallel overexpression of RBFOX RBPs, as reintroducing RBFOX1 in GBM cell lines was shown to inhibit tumorigenesis and its knock-down was demonstrated to compromise neuronal lineage differentiation of premalignant neural stem cells[57]. Targeting of mRNAs and mRNA-RBP interactions with antisense oligonucleotides[58,59] or small molecules[60] holds much promise for medical applications. As MAPP is a fully automated workflow, the task of identifying regulators of pre-mRNA processing from RNA-Seq datasets is considerably facilitated.

Other groups have investigated binding site location-dependent effects of RBPs, specifically proposing the concept of "RNA maps"[61], which summarize the density of RBP binding sites in the vicinity of various types of landmarks (exon and transcript boundaries), where RBPs exert regulatory roles. For instance, binding of the Nova RBP upstream of a cassette exon is associated with the skipping of that exon, while the binding downstream of the cassette exon is associated with the exon inclusion. The impact maps that MAPP constructs provide complementary information. They do not rely on direct information about the location of the binding site of the RBP (usually obtained with CLIP) nor on specific thresholds for defining regulated events such as exon inclusions. Rather, MAPP makes use of the quantitative information in the inclusion level of each exon or PAS as well as in the number of predicted binding sites in the vicinity of these exons. As a result, MAPP provides quantitative information about the impact of motifs on RNA processing, circumventing issues regarding the coverage of the binding sites by CLIP in targets with different levels of expression. Also interesting to note is the increasing use of massively parallel assays for exploring the dependence of RNA processing on specific motifs[62]. These provide information more analogous to MAPP's impact maps, but are limited to a small number of conditions, a small number of targets, and regions within these targets, and have been so far used to characterize general principles of RNA processing. In contrast, MAPP's utility comes primarily in exploring a broad range

of conditions and identifying condition/tissue-specific regulators. Thus, MAPP extends the RNA biologist's toolbox to enable the functional characterization of RBP-RNA interactions and their consequences at a high level of detail.

In conclusion, we developed a powerful computational approach to identify regulators of splicing and 3′ end processing, which are frequently coordinated. MAPP has been developed using modern principles of high-quality scientific software engineering, facilitating further development by a broad community of developers.

## Methods

### Datasets

We validated our method on publicly available RNA-Seq data with perturbed levels of RBPs with known impact on splicing, and, to some extent, polyadenylation. We have also applied MAPP to other publicly available datasets related to brain malignancies, normal brain tissue samples, a neural stem cell model, human fetal neurocortical tissue, CFIm knock-down and overexpression, and a microRNA mimic transfection. The full list of samples' and records' IDs is included in Supplementary Data 1. Similarly, Supplementary Data 2 lists all RNA-Seq datasets related to RBP knock-downs we have obtained from the ENCODE project. To apply MAPP we require that samples meet minimal criteria of quality. For example, we require a sufficiently high Transcript Integrity Number (>50, typically >70)[63], high proportion of uniquely mapped reads (>0.95), high proportion of high-quality mapped reads (>0.85), low level of rRNA contamination (<0.05) and low proportion of reads mapped to intergenic regions (<0.1), as reported by RNA-SeQC[64]. BAM files with mapped RNA-Seq reads of normal and tumor sample pairs from TCGA were obtained from the Genomic Data Commons (GDC) data portal[65]. The selection of normal-tumor pairs from glioblastoma data was done as described previously (Supplementary Data 3)[6]. Additional transcriptomic alignments were generated by first unmapping and then re-aligning RNA-Seq reads, utilizing Samtools and STAR with proper command line options.

### MAPP

MAPP, standing for motif activity on pre-mRNA processing, is implemented as a modular snakemake workflow[66] with distinct standalone sub-workflows dedicated to separate functionalities. These are RNA-Seq data preprocessing, selection of cassette exons, selection of tandem poly(A) sites, quantification of exon inclusion, quantification of poly(A) site usage, generation of motif count matrices (PWMs/k-mers) in each window around each site, the MAEI model (splicing), the KAPACv2.0 model (polyadenylation), and the summary of results. Each of these modules is described in detail in the Supplementary Methods. MAPP supports two distinct software technologies: Conda environments[67] and Singularity containers[68].

### MAEI

MAEI, which stands for motif activity on exon inclusion, is a model designed to infer the impact of short sequence motifs on the differential inclusion of cassette exons. In order to prevent the confounding effect of sites located within intronic regions, by default, MAPP fits activities for windows of 50 nt length and considering only exons that are at least 50 nt in length. As input, the MAEI model uses, for each exon e, the expression levels of transcripts including and excluding the exon across a set of samples s, together with a matrix N whose entries $N_{e,m}$ correspond to the motif counts of each motif m in a window around the mRNA processing sites of interest, i.e., 5′SS or 3′SS, for each exon e. The motifs can either be specified as PWMs or k-mers. We model the inclusion fractions $f_{e,s}$ (i.e., the fraction of transcripts including the cassette exon e among transcripts for which e was included in the pre-mRNA, see Supplementary Methods) of every exon e in every sample s using a logistic function: $\Theta_{e,s} = \frac{e^{X_{e,s}}}{1 + e^{X_{e,s}}}$, where $X_{e,s} = b_s + c_e + N_{e,m} * A_{m,s}$ is a linear function of the model parameters:

$b_s$—the baseline inclusion rate of all exons in sample s, $c_e$—the baseline inclusion rate of exon e in all samples and $A_{m,s}$—the "activity" of a motif m in a sample s. $N_{e,m}$ denotes the number of binding sites of motif m in the proximity of exon e, which is either given by the sum of site probabilities predicted with the PWM or the raw k-mer counts). We fit this logistic regression model using a Bayesian approach resulting in inferred motif activities $A_{m,s}$ with corresponding error bars $\sigma_{m,s}$ and finally obtain for each motif m in sample s a z-score $Z_{m,s} = \frac{A_{m,s}}{\sigma_{m,s}}$.

The activity z-scores are then presented visually on the impact maps. See the Supplementary Methods for more details on all calculations.

In order to distinguish motifs with statistically significant z-scores from those with z-scores from a Gaussian background distribution we use a Gaussian mixture model to renormalize the z-scores and transform them into p values from a standard normal distribution. Statistical significance is then finally assessed upon Bonferroni-correction of these p values. Again, we refer the reader to the Supplementary Methods for more details on the procedure.

### KAPACv2.0

KAPACv2.0, standing for K-mer Activity on PolyAdenylation site Choice version 2.0, implements a more general version of our previously published KAPAC tool[6]. KAPACv2.0 models genome-scale changes in 3′ end usage to infer sequence motifs that can explain 3′ end site usage across samples. In contrast to KAPAC, KAPACv2.0 can use both binding sites predicted with position-dependent weight matrices (PWMs) as well as k-mer counts. Also, while the first version of KAPAC was designed to run on sample contrasts, such as knock-down versus control samples, KAPACv2.0 does not require contrasts but can be applied to any set of samples, such as different tissues or a time series. First, we define the relative usage of poly(A) site p in sample s as $u_{p,s}$. KAPACv2.0 then models the relative usage $u_{p,s}$ with respect to the mean of all samples as a linear function of the occurrence of PWM binding sites or k-mer counts and the unknown "activity" of these PWMs/k-mers: $\log_2\left(u_{p,s}\right) = N_{p,k} * A_{k,s} + c_p + c_{s,e} + \varepsilon$, where $N_{p,k}$ is the number of binding sites (predicted with the PWM or by k-mer counting) around poly(A) site p, $c_p$ is the mean $\log_2$ relative usage of poly(A) site p across all samples, $c_{s,e}$ is the mean $\log_2$ relative usage of the poly(A) site from exon e in sample s and ε is the residual error. Finally, $A_{k,s}$ is the activity of the PWM/k-mer k in sample s, which determines how much the PWM/k-mer contributes to the relative usage of the poly(A) site. KAPACv2.0 calculates for each sample s and every PWM or k-mer k, respectively, a z-score $Z_{k,s} = A_{k,s}/\sigma_{k,s}$, whereas $\sigma_{k,s}$ are the fitting errors of the activities $A_{k,s}$. Background correction and ranking of PWMs/k-mers is done as described for the MAEI approach above (see Supplementary Methods and ref. 6 for further information).

### Curation of PWMs of RBPs binding motifs

ATtRACT is a publicly available database of RNA-binding proteins and associated motifs[69]. On 20 August 2021 we downloaded the zip file containing all available RBP motifs in the format of position-dependent weight matrices (giving the probability of observing any of the four bases at each position of the binding site) as well as their corresponding metadata (ATtRACT_db.txt). From the ATtRACT_db.txt we first selected motifs annotated with the species *Homo sapiens* (3256 records) and from these only those that corresponded to wild-type proteins ("Mutated" field having the value "no", 3178 records). We next selected only one of the records that had the same gene ID, PWM ID and experiment description (where for experiment description, records that contained the word 'SELEX' were considered as having the same description). This procedure resulted in 1120 records. Next, we clustered records for which the entries in the PWMs (position-dependent frequencies of nucleotide occurrence) were identical. If the cluster with identical PWM entries contained multiple RBPs, we discarded them all, as we could not unambiguously assign the PWM to

one RBP. If the cluster contained multiple records for the same RBP, we kept only one of them. This step left 523 records. We further determined the length of the core motif for each PWM, that is, the longest motif, such that the first and last positions had a non-zero information content and discarded those records where core motifs were not in the range of 4 to 7 nucleotides. This step left 346 records. Finally, for each PWM, we calculated the total motif entropy and discarded those that were too degenerate (with an entropy higher than 10) using functions adapted from the SMEAGOL toolbox[70]. This procedure yielded 344 PWMs for the MAPP analyses.

## Coverage profiles of RNA-binding proteins

To gain additional confidence in MAPP's inferences, we constructed coverage profiles for distinct RNA-binding proteins based on eCLIP data in HepG2 and K562 cells, publicly available as a part of the ENCODE project (experiment IDs: ENCSR550DVK, ENCSR987FTF, ENCSR384KAN, ENCSR249ROI, ENCSR756CKJ, and ENCSR981WKN). For Figs. 2a, b, 4a we have used the experiments conducted in HepG2 cells (similar plots for K562 cells can be found in Supplementary Figs. S3, S7) we selected the group of the top 200 targets with the highest change in alternative splicing as well as alternative poly-adenylation into the expected direction based on the average quantified exon inclusion fraction and poly(A) site usage, respectively. We have extended the margins around these sites so that the eCLIP analysis matches the regions covered by our MAPP sliding windows. For every RNA processing site separately, we have calculated foreground/background ratios of library-size-normalized position-wise eCLIP read coverages (foreground being eCLIP reads from the RBP pulldown experiment and background being the corresponding control pulldown experiment). We have plotted the position-wise mean ratio over all sites (smoothened by the −5/+5 nt of each position). Additionally to the target set we selected a group of 1,000 sites with the least change in RNA processing and treated them as background to estimate coverage profiles for non-targets. From this set, there were randomly sampled 200 non-targets for 100 times, each time following the same procedure as described above for the non-random sites in order to obtain 100 background coverage profiles. These random profiles were used to plot the (smoothened) per-position mean of means together with a confidence boundary which reflects the per-position standard deviation of the means. The data processing notebook is available in the supplementary data (see Data availability section).

## Selection of ENCODE experiments, reported motifs, and k-mers

We have downloaded and analyzed RNA-Seq samples linked to 472 knock-down experiments of RBPs, publicly available as a part of the ENCODE project; 16 of these did not pass the quality-control step as defined in the "Datasets" section. We used the remaining 456 datasets for the analysis shown in Fig. 3. Briefly, each ENCODE experiment has been analyzed with MAPP in both PWM-based and k-mer-based approaches. For each of the knock-down experiments, we selected the lowest rank of any ATtRACT PWM associated with the perturbed RBP for which MAPP found a statistically significant impact on any of the signals (statistical significance annotated with the "abs" strategy, please see Supplementary Methods). Such obtained "PWM rank" is the key by which the results table is sorted in descending order. For only 12 ENCODE experiments, we found that the PWM associated with the perturbed RBP had a rank of a maximum of 5 (out of 344 curated PWMs). For these, we report the PWM ranks and their impact maps, as inferred by MAPP. We then checked whether the appropriate motif is also recovered in the k-mer mode. For this, we selected the k-mer with the highest overall statistically significant activity z-score, averaged over all of the processing sites (labeled as "1st ranked k-mer") from each experiment. Alongside the previously described PWM-based results, we report the 1st ranked k-mer and its impact map. Data

processing scripts are available in the supplementary data (see Data availability section).

## Reporting summary

Further information on research design is available in the Nature Portfolio Reporting Summary linked to this article.

## Data availability

The results generated in this study have been deposited in the Zenodo database under accession code 5789986[71]. The raw RNA-Seq data are accessible at ENCODE[21] and the following records: GSE56010, GSE69656, GSE71468, GSE179630, PRJNA798408, GSE204705, GSE185861, GSE147352, GSE38805, PRJEB4337, All accession numbers and sample ids are available from Supplementary Data 1, 2, and 3.

## Code availability

The MAPP code is available on GitHub (https://github.com/gruber-sciencelab/MAPP) and in the Zenodo database under accession code 10845501[72].

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

## Acknowledgements

We express our wholehearted gratitude to the sciCORE facility of the University of Basel and to Stefan Gerlach from the Scientific Compute Cluster of the University of Konstanz (SCCKN) for maintaining the high-performance computing clusters where we have carried out all the computations and analyses. This work was supported by the Deutsche Forschungsgemeinschaft (grant GR 6074/3-1, 517370297 to A.J.G.) and the Swiss National Science Foundation (grant 310030_189063 to M.Z.). M.B. was a recipient of a PhD scholarship "Fellowship for Excellence" of the Biozentrum, University of Basel. The results published here are in part based upon data generated by The Cancer Genome Atlas managed by the NCI and NHGRI. Information about TCGA can be found at http://cancergenome.nih.gov.

## Author contributions

M.B. and A.J.G. designed the MAPP pipeline, M.B. implemented the pipeline with contributions from A.J.G. I.U.K. helped testing the pipeline and M.B. prepared the GitHub repository. E.v.N. and M.B. designed the MAEI model and background correction and M.B. implemented it. A.J.G. designed and implemented the KAPACv2.0 model. R.S. developed the TPA module of the MAPP pipeline. M.B., A.J.G., I.U.K., and T.G. analyzed the data. M.B. created the supplementary data record. A.J.G. conceived the study. A.J.G. and M.Z. designed and supervised the project. A.J.G., M.B., and M.Z wrote the manuscript.

## Funding

## Competing interests

The authors declare no competing interests.
