## [Peer Review File · Nature Communications]

MAPP unravels frequent co-regulation of splicing and polyadenylation by RNA-binding proteins and their dysregulation in cancerREVIEWER COMMENTS

Reviewer #1 (Remarks to the Author):

This manuscript by Bak et al. reports a bioinformatics analysis method named MAPP, which combines a previously developed method for alternative polyadenylation (APA) analysis (KAPAC) with a new method for alternative splicing (AS) analysis of cassette exons (MAEI). The method predicts motifs for RNA-binding proteins (RBPs) and their positions in regulation of AS and/or APA. The authors first tested MAPP on >400 RBP knockdown (KD) data sets generated by ENCODE labs, which allowed them to compare their motif predictions with RBP-binding sites identified by eCLIP. They reported that some RBPs have dual functions in AS and APA. They confirmed their previous finding of hnRNPC in AS and APA regulation and indicated that hnRNPK and PCBP1 have distinct regulatory modes despite that both are C-rich motif RBPs. They then applied MAPP to analyze the transcriptome data of glioblastoma. They found that PTBP1 and RBFOX coordinately drive the oncogenic splicing program in glioblastoma, which is consistent with previous studies. Overall the work was well carried out and the program seems useful in identifying RBP-mediated AS and APA. However, because this is a bioinformatics/data mining study, the manuscript would be much strengthened and can have a broader readership if more diseases or biological conditions are examined.

Major:

1. Regarding APA regulation by hnRNPK and PCBP1, because both bind upstream 3'UTR sequences, can the authors rule out the possibility that their binding regulates 3'UTR isoform stability rather than polyA site usage?
2. Of the 400 RBPs studied by ENCODE, some are considered core 3' end processing factors. Do the authors find 3' end processing factors have regulatory roles in AS? This could be quite informative to the field.
3. The GBM analysis is largely confirmatory of their previous data. The manuscript would be much strengthened if other diseases are also examined.

Reviewer #2 (Remarks to the Author):

In this study, Bak et al developed a computational tool named MAPP to identify cis-regulatory elements regulating alternative splicing (AS) and polyadenylation (APA) using RNA-seq. And then they applied this method to the RNA-seq datasets from different experimental conditions, including the knocking down of different RNA binding proteins (RBPs) and brain cancers. While the authors showed some useful cases applying this method, I have some major concerns and questions about their method as well as the conclusions. I feel this study does not have significant advances in computational methods or biological discoveries to be published in Nature Communications. Following are detailed comments.

1. The general computational approach the authors used to identify cis-regulatory elements was published before (PMID: 29592812). This manuscript represents a modest improvement vs. their previously published software.
2. I have some major concerns about the computational approach the authors proposed. To infer the motifs contributing to the regulation of AS and APA, the authors included all the cassette exons and polyA sites in the analyses. The manuscript did not describe any pre-filtering steps to ensure the exons, polyA sites or associated genes are well expressed with enough sequencing read coverage. The lowly expressed events may lead to noises in calculating the exon usage and polyA site expression levels.
3. The authors claimed that they did not select significant regulated AS or APA events to identify their

regulatory motifs. Instead, they included all the database-annotated events which different regulatory levels to train the model. But, during most biological processes, only a small fraction of AS and APA events show significant regulation or have enough read coverage to reliably infer their expression levels. The general approach the authors proposed probably cannot be applied to many biological conditions.

4. The software mainly outputs the meta plots of motifs and their significance scores around exons and polyA sites. It would be better if the software can output significantly regulated AS and APA events as well as the motif existence for each event, which will be helpful for experimental biologicals to find interesting genes to follow up.

5. The Method section tends to lack critical detailed information about the data analysis steps. They described the concept of the models but without any detailed information about data quality controls or parameters they used for the analyses of individual samples. For example, how did not they do the read mapping? How did they ensure the genes, exons or polyA sites are well expressed? It is hard to judge the rigor of their analyses.

6. Another major concern I have is that the manuscript did not describe solid novel biological findings. The motifs associated with the factor knockdowns were limited to the factor itself. An advantage of the approach is to study the interplay between different RBPs. The authors may discover some novel interplay in the RBP knockdown datasets.

7. A novelty seems to be the finding of PTBP1 and RBFOX motifs in Glioblastoma. The results were limited to the correlation of gene expression and regulatory events. It is unclear whether PTBP1 and RBFOX as well as their interactions are indeed important to Glioblastoma.

8. For the analyses of glioblastoma samples, they listed 5 healthy and 5 tumor samples in the supplementary tables. It is unclear how these 10 samples were selected from the hundreds of clinically available samples. Will the authors' conclusions still be valid if they apply their analyses to all samples?

Reviewer #1 (Remarks to the Author):

This manuscript by Bak et al. reports a bioinformatics analysis method named MAPP, which combines a previously developed method for alternative polyadenylation (APA) analysis (KAPAC) with a new method for alternative splicing (AS) analysis of cassette exons (MAEI). The method predicts motifs for RNA-binding proteins (RBPs) and their positions in regulation of AS and/or APA. The authors first tested MAPP on >400 RBP knockdown (KD) data sets generated by ENCODE labs, which allowed them to compare their motif predictions with RBP-binding sites identified by eCLIP. They reported that some RBPs have dual functions in AS and APA. They confirmed their previous finding of hnRNPC in AS and APA regulation and indicated that hnRNPK and PCBP1 have distinct regulatory modes despite that both are C-rich motif RBPs. They then applied MAPP to analyze the transcriptome data of glioblastoma. They found that PTBP1 and RBFOX coordinately drive the oncogenic splicing program in glioblastoma, which is consistent with previous studies. Overall the work was well carried out and the program seems useful in identifying RBP-mediated AS and APA. However, because this is a bioinformatics/data mining study, the manuscript would be much strengthened and can have a broader readership if more diseases or biological conditions are examined.

First, we would like to thank the reviewer for these positive words and for pointing out that our work was well carried out and that our novel MAPP approach is useful for identifying RBPs involved in alternative splicing and polyadenylation. We also appreciate the suggestion by the reviewer to extend our manuscript with further datasets.

In our revised and largely extended manuscript we are now presenting the analysis of additional 321 RNA-seq libraries from 7 new datasets, resulting in one entirely new figure in the main manuscript (**Manuscript Figure 6**, comprising of 8 new subfigures), plus additional 10 data analysis Supplementary Figures. A detailed list of all analysed RNA-seq samples is now provided in Supplementary Tables S1-3. Further details about our novel analyses can be found in our detailed response to the reviewers comment number 3 below.

Major:

1. Regarding APA regulation by hnRNPK and PCBP1, because both bind upstream 3'UTR sequences, can the authors rule out the possibility that their binding regulates 3'UTR isoform stability rather than polyA site usage?

The reviewer raises an important point about the PCBP1 and HNRNPK RNA-binding proteins, for which MAPP has inferred that they impact cleavage and polyadenylation by binding upstream of the cleavage and polyadenylation sites (**Manuscript Fig. 3b**).

We have now performed careful additional analysis in order to address the reviewer's question. To determine whether HNRNPK or PCBP1, respectively, regulate mRNA stability we have taken an analysis approach that appears to be most suitable to disentangle the effects potentially stemming from mRNA degradation and alternative cleavage and polyadenylation (**Figure 1.1, below**; all analysis details are provided in Supplementary Methods Section 2.11).

First, to exclude confounding effects stemming from alternative cleavage and polyadenylation (APA) we only considered terminal exons (TEs) of mRNAs having exactly one cleavage and polyadenylation (CPA) site, i.e. we excluded all terminal exons that overlap with any other terminal exon in the current ENSEMBL gene annotation (**Figure 1.1a**). Since we know from our previous research that transcript 3' ends are poorly annotated, we have further extended our approach by incorporating experimental evidence from 3' end sequencing data collected and curated in our PolyASite atlas (Gruber et al., 2016, *Genome Research*; Hermann et al., 2021, *Nucleic Acids Research*). That is, to reduce false positives in our downstream analysis we have dropped TEs that do not intersect with an experimentally observed transcript 3' end (**Figure 1.1b**). Further, to exclude that our analysis is confounded by not yet annotated TEs, we have also removed TEs that intersect with more than one experimentally observed transcript 3' end (**Figure 1.1c**).

Figure 1.1: Approach used to analyse a potential impact of the HNRNPK RBP and the PCBP1 RBP on mRNA stability, respectively.

To exclude that our stability analysis is impacted by alternative splicing, we only retained TEs that contain a Stop codon (**Figure 1.1d**). Finally, as it is known that *cis*-regulatory elements that impact mRNA stability are typically located within the 3' untranslated region (3' UTR), for our stability analysis we have counted the investigated sequence elements only within the 3'UTRs, but not coding sequences (CDS) (**Figure 1.1e**). To determine the impact of the investigated sequence elements on mRNA stability, we have calculated the log₂ fold-change of the TE-corresponding transcript expression upon knock-down of HNRNPK and PCBP1, respectively. Finally, we determined whether TE-containing transcripts having the MAPP inferred sequence element ('CCC') in their 3' UTR have significantly larger or smaller log₂ fold-changes, respectively, compared to transcripts being associated with TEs that have no 'CCC' sequence element in their 3' UTR (**Figure 1.1f**). In summary, within our approach we

use only TEs that should not be impacted by alternative cleavage and polyadenylation as they have exactly one experimentally observed transcript 3' end and no other TEs overlap with their genomic coordinates. Also the expression of transcripts including these TEs should not be impacted by the alternative presence/absence of stability-regulating cis-regulatory elements as 3' UTR shortening/lengthening can be excluded. As positive control we have applied our approach to a microRNA (miR) overexpression experiment (hsa-miR-130-3p), but counting its seed sequence ('TTGCACT') in the 3' UTRs. The obtained results demonstrate that our approach is able to properly capture the impact of 3' UTR binding regulators on mRNA stability, as it consistently infers a significant impact of the microRNA for all three overexpression experiments (**Figure 1.2**). Importantly, as expected for a microRNA our approach consistently infers a destabilizing effect (3' UTRs containing sites exhibit in all three cases significantly more negative log₂ fold-changes compared to 3'UTRs having no sites).

Figure 1.2 (new Supplementary Fig. S15): Effect of a microRNA (hsa-miR-130-3p) overexpression on mRNA stability as inferred from three replicate experiments. p: p-value obtained by applying a two-sided Wilcoxon rank-sum test to the log₂ expression fold-changes of transcripts with or without binding sites for the microRNA.

Figure 1.3 (new Supplementary Fig. S15): Analysis of the impact of HNRNPK and PCBP1 on mRNA stability, respectively.

Applying the above described approach to the PCBP1 and HNRNPK knock-down datasets shows that neither the PCBP1 RBP nor the HNRNPK RBP has a significant effect on mRNA stability (**Figure 1.3, above**), with the exception of replicate 1 of the second HNRNPK knock-down experiment (ENCSR529JNJ). However, given that HNRNPK has a not significant trend towards stabilizing mRNAs in both replicates of the first experiment (ENCSR853ZJS) and the opposite trend (towards destabilizing mRNAs) in the second experiment (ENCSR529JNJ), the one outlier replicate of the ENCSR529JNJ experiment misses out significant reproducibility and is further contradicted by opposing trends, though not significant, observed in the ENCSR853ZJS experiment.

The statistics of our analysis speaks against a pronounced effect of HNRNPK or PCBP1 that could have caused the significant effect on 3' end processing that MAPP has inferred for the two RBPs. Also, a role in 3' end processing fits to the localization of the RBPs in the nucleus, whereas they appear to also shuttle to other cell compartments (e.g. reported by the Human Protein Atlas, <https://www.proteinatlas.org/ENSG00000165119-HNRNPK/subcellular>; <https://www.proteinatlas.org/ENSG00000169564-PCBP1/subcellular>).

Finally, we would like to also mention that MAPP is in fact stable against confounding by mRNA stability alterations as it operates on relative usages of poly(A) sites rather than transcript abundance and it fits activities for specific distances relative to the cleavage sites. However, we fully agree with the reviewer that this was not yet obvious from the materials we had provided. Thus, we have now also added a MAPP run on the microRNA overexpression experiment to our revised manuscript, which demonstrates that MAPP infers no significant 3' end processing impact for the microRNA binding sequence (**Figure 1.4 below**; new **Supplementary Fig. S16**). Please note that for the same binding sequence (TTGCACT) and the same set of microRNA overexpression experiments our mRNA stability analysis approach consistently infers a significant destabilizing effect of the microRNA on mRNAs (**Figure 1.2, above**).

Figure 1.4 (new Supplementary Fig. S16): Impact map for the hsa-miR-130-3p binding sequence (TTGCACT) inferred with MAPP from the samples of a hsa-miR-130-3p overexpression dataset (Accession Number: GSE204705).

As we believe that the reviewer has asked here a very relevant and interesting question that might be of broad interest to the readership, we have now integrated this analysis into our revised manuscript (**Supplementary Methods Section 2.11 & Supplementary Figure S15,S16**).

2. Of the 400 RBPs studied by ENCODE, some are considered core 3' end processing factors. Do the authors find 3' end processing factors have regulatory roles in AS? This could be quite informative to the field.

As the reviewer has pointed out, the ENCODE dataset contains RBPs that have previously been reported to act on 3' end processing. Some are core factors (CPSF6, CPSF7, CSTF2, CSTF2T), while others are not, though they have been linked to 3' end processing before (PCBP1, HNRNPC, HNRNPK). For the latter, running MAPP in k-mer mode in order to cover the full sequence space, we recovered the previously reported sequence motifs and impact on 3' end processing (**Figure 1.5a, below**). Importantly, MAPP has also inferred the known function of these RBPs in regulating splicing by acting on similar sequence motifs located in the vicinity of 3'SS and/or 5'SS (**Manuscript Fig. 3a**).

Figure 1.5: MAPP impact maps from knockdown experiments of (a) ENCODE RBPs that are known to act on 3' end processing, (b) ENCODE 3' end core factors, and (c) 3' end core factors from publicly available datasets (new **Supplementary Fig. S4**).

However, for CPSF6 (CFIm68), CPSF7 (CFIm59), and CSTF2 (CSTF-64) MAPP inferred only very minor signals for the known binding motifs (**Figure 1.5b, above**). The reasons are probably multiple, ranging from a low efficiency of knock-down, to compensatory effects (e.g. CSTF2/CSTF2T). Since we know from our previous research that these factors (except CPSF7) do have systematic effects on 3' end processing (reviewed in Gruber & Zavolan, 2019, *Nature Reviews Genetics*; PMID: 31267064, multiple references therein) we have supplemented the ENCODE data with other publicly available knock-down datasets for the CFIm68 and CFIm25 factors. The MAPP runs showed the expected activating impact of the known CFIm binding sequence, UGUA, on poly(A) site processing upstream from the poly(A) signal, i.e. upstream of -25 nt from the cleavage site (**Figure 1.5c, above**), but they did not reveal any impact on splicing (**Figure 1.5c, above**). As we believe that these results are indeed very interesting and greatly add to our manuscript, we have now also included a discussion of these results within our revised manuscript (page 4 & new **Supplementary Fig. S4**).

3. The GBM analysis is largely confirmatory of their previous data. The manuscript would be much strengthened if other diseases are also examined.

We thank the reviewer for the suggestion, and we are sure that MAPP will be applied to other diseases and experimental datasets by the research community. In this manuscript our aim was to provide a more complete, accurate and automated analysis not only of 3' end processing, but also of splicing, to be able to detect dual activities of RBPs, which do indeed appear to be common. Thus, in contrast to our previous work, we introduce a completely novel computational modelling approach, MAEI, plus an improved 3' end processing model and a unified background correction strategy. For GBM we recover prior results, but the analysis is much expanded. To further follow the reviewer's recommendation as well as the Editors request to e.g. add analyses on 3' end processing factors and more biological/disease conditions, we have now added analyses of 3' end processing factors (see our answer to question 2 above) and of additional biological conditions. As Reviewer 2 had requested to provide additional glioblastoma data, and also in the light of the story-line of our manuscript, it made most sense to focus in this revision on this disease, but expanding the analysis to other aspects of it and with additional datasets. In particular, we have analysed further glioblastomas, but also data from other brain tumours and tissues as well as neural stem cells, thereby expanding our findings towards the origins of glioblastomas. These efforts resulted in an entirely new figure in our revised manuscript (**Manuscript Figure 6**) plus 5 new Supplementary Figures (**S8,S9,S11,S12,S13**).

Below we list the conditions and datasets that we have additionally analysed and which present now in our vastly extended and revised manuscript (all samples are comprehensively listed in **Supplementary Table S1**):

1. Neural Stem Cell Model (9 samples)
2. Ventricular zone residing cells (6 samples)
3. Cortical plate residing cells (6 samples)
4. Oligodendroglioma (4 samples)
5. Astrocytoma (2 samples)
6. Glioblastoma (99 samples from 2 datasets)
7. Normal brain (15 samples from 1 datasets)
8. Human tissues (160 samples)
9. CFIm25 knockdown (8 samples)
10. CFIm68 knockdown (9 samples)
11. microRNA (hsa-miR-130-3p) overexpression and controls (6 samples)

As outlined above, we added a total of 321 additional RNA-seq libraries to our study, bringing in the 11 biological/experimental conditions mentioned above, 9 of which are novel for our revised manuscript. We hope that the reviewer agrees that these novel datasets valuably extend our manuscript at various places and elegantly extend our story-line going from RBP knockdown experiments, to glioblastoma cells and their splicing similarity to neuronal progenitors.

Finally, we would like to thank the reviewer once again for all the valuable suggestions and for carefully reading and considering our vastly extended analysis and revised manuscript.

Reviewer #2 (Remarks to the Author):

In this study, Bak et al developed a computational tool named MAPP to identify cis-regulatory elements regulating alternative splicing (AS) and polyadenylation (APA) using RNA-seq. And then they applied this method to the RNA-seq datasets from different experimental conditions, including the knocking down of different RNA binding proteins (RBPs) and brain cancers. While the authors showed some useful cases applying this method, I have some major concerns and questions about their method as well as the conclusions. I feel this study does not have significant advances in computational methods or biological discoveries to be published in Nature Communications. Following are detailed comments.

1. The general computational approach the authors used to identify cis-regulatory elements was published before (PMID: 29592812). This manuscript represents a modest improvement vs. their previously published software.

We acknowledge the thoughts of the reviewer concerning the computational approach presented here, but would like to kindly emphasise that it is not the case that MAPP represents only a modest improvement of our previous work.

As discussed in one of our review articles, the pre-mRNA processing field tries to answer the question of the co-regulation of alternative splicing and polyadenylation for a long time (Gruber & Zavolan, 2019, *Nature Reviews Genetics*; PMID: 31267064). Also, in prior work we have uncovered that alternative splicing and alternative polyadenylation give rise to an abundance of so far unknown mRNA isoforms (Gruber et al. 2018, *Nature Methods*; PMID: 30202060). However, in order to characterize the RNA-binding proteins (RBPs) that regulate the processing of specific mRNA isoforms, data from crosslinking and immunoprecipitation (CLIP) or comparable experiments are required. However, CLIP experiments are only available for a small minority of the ~1,500 RBPs present in human. Taking into account the expenditure of conducting paired CLIP & RNA-seq experiments in any system of interest and the current pace in the field to create such data, it will still take a very long time before there will exist a comprehensive characterization of the regulatory cross-talk of alternative splicing and polyadenylation acting in healthy and diseased cellular states.

Our motivation behind developing MAPP was to enable the automated inference of both the binding specificities and the functional impact of RBPs on mRNA maturation solely from standard RNA sequencing (RNA-seq) data in order to enable the community to study pre-mRNA processing in any system of interest for which RNA-seq data are available.

Importantly, while for many questions about the regulation of gene expression there exist dozens of bioinformatics solutions, MAPP represents the first computational approach able to uncover co-regulation of alternative splicing and polyadenylation from RNA-seq data. MAPP makes use of a completely novel quantitative model of splicing (MAEI), plus a vastly improved and generalised 3' end processing model and applies carefully developed statistical measures to combine both models towards unified quantitative maps that provide detailed insights on the crosstalk of splicing and 3' end processing in any system of interest.

In summary, while KAPAC from our 2018 study (PMID: 29592812) implemented only the final step of the inference of RBP activities on polyadenylation in ~1.5k lines of R code, MAPP comprises of a fully integrated and automated pipeline that combines >50 bioinformatics tools and tailored scripts, spanning tens of thousands of lines of code developed to enable automated quality control and the analysis of RNA-seq data, from the raw fastq-sequencing files to the final MAPP summary report (**Supplementary Figures S1 and S2**). All these features, and most importantly the development of the new modelling approaches as well as their thorough validation and integration, took a great amount of time and effort, finally delivering the first computational end-to-end solution for studying the concerted impact of RBPs on pre-mRNA processing. Thus, we feel it is important to highlight that MAPP is far from being a modest improvement of previous work.

Finally, we would like to mention that in revising our manuscript, we have also updated the MAPP software following the reviewers' suggestions (please see our answers below) and we are presenting the analysis of 9 additional biological/experimental conditions (321 additional RNA-seq libraries in total), thereby extending various aspects of our manuscript, including our previous story-line from RBP knockdown experiments, to glioblastomas and now adding data that highlight their similarity to neuronal progenitor cells within an entirely new figure in our revised manuscript (**Figure 2.7, below; Manuscript Figure 6**).

2. I have some major concerns about the computational approach the authors proposed. To infer the motifs contributing to the regulation of AS and APA, the authors included all the cassette exons and polyA sites in the analyses. The manuscript did not describe any pre-filtering steps to ensure the exons, polyA sites or associated genes are well expressed with enough sequencing read coverage. The lowly expressed events may lead to noises in calculating the exon usage and polyA site expression levels.

We would like to thank the reviewer for bringing up this very important point. Indeed it is crucial to appropriately handle the noise in the data to enable the inference of meaningful signals from the genome-scale measurements. For this reason, over the past years we have spent a lot of time and effort to develop tools and enhance gene annotations, to finally be able to capture the impact of global regulators. Below we describe the most important steps implemented in MAPP for quantifying and filtering the sets of cassette exons and 3' end processing events, respectively, to increase the quality of the event sets ultimately used by our models:

- **Sample quality control:** MAPP has a dedicated "Preprocessing and Quality Analysis" (short PQA) module. (**Figure 2.4, below**), which determines an array of quality metrics for RNA-Seq data relevant for our models and automatically removes samples (unless turned off by the user) that do not fulfil the minimum quality criteria required for a high quality MAPP run. We describe the PQA module in more detail in response to the reviewers question number 5 (below) and in our updated and much improved Supplementary Methods Section "2.1 Data preprocessing and quality analysis".
- **Selection of cassette exon events:** MAPP has a dedicated "Alternative Spliced Exons" (short ASE) module (**Figure 2.2a, below; Supplementary Methods Section**

2.2), which makes use of 7 tools and tailored scripts to (i) generate a set of potential alternative cassette exon events (based on the SUPPA2 tool, PMID: 29571299), (ii) filter the events by the minimal/maximal exon lengths to ensure that the inferred MAPP activities located within exons are not confounded by intronic sequence elements as well as by the proximity to chromosome boundaries, (iii) cluster largely overlapping cassette exon events based on their mutual coverage in order to prevent duplication of processing sites, and finally (iv) select representative events of each cluster: those supported by the greatest number of distinct transcripts. For the quantification of exon inclusion MAPP makes use of its “Quantification of Exon Inclusion” (short QEI) module (Figure 2.2b) which utilises the Salmon tool (PMID: 28263959) to quantify transcripts’ expression levels that are later aggregated to reflect an exon-centric information. Importantly, we keep only cassette exons originating from genes expressed at least 1 TPM in all samples (please also see our response to the reviewers question 5 below). In brief, a TPM below 1 wouldn’t be feasible, as MAPP is operating on changes, which can be only observed in both directions, e.g. more/less inclusion, if there is a minimum basal expression guaranteed. However, a TPM greater than 1 will filter out events that might or might not be informative. As the reviewer correctly pointed out: Setting the threshold too low might result in the inclusion of noisy events, whereas setting the threshold too high will remove true positive events and thus decrease the statistical power available for our models. To find out which threshold is best suited for most datasets we have benchmarked the performance of MAPP based on our 3 validation datasets (HNRNPC knock-down, RBFOX overexpression and PTBP1/2 double knock-down). Running MAPP with a TPM filter ranging from 1 to 15 for each of the datasets allowed us to empirically determine that a TPM threshold of 1 delivers for all datasets the most significant results (Figure 2.1, below). Thus, we use this threshold as default within our MAPP pipeline. Nevertheless, this threshold can be manually adapted by expert users in order to e.g. perform a benchmarking for a specific dataset by setting the *min_transcript_expression* parameter in the MAPP config file (see Supplementary Methods Section 2.7).

Figure 2.1 (new Supplementary Fig. S14): MAPP reported motif activity z-scores of the top ranked motifs acting around its respective sites as a function of minimum TPM cutoff for expressed genes to be considered in MAEI: (a) HNRNPC knock-down (motif TTTTT acting on 3'SS), (b) PTBP1/2 knock-down (motif TCT acting on 3'SS), and (c) RBFOX1 overexpression (motif GCATG acting on 5'SS).

- Selection of poly(A) sites:** MAPP has a dedicated “Tandem Poly(A) site” (short TPA) module (**Figure 2.2c, below; Supplementary Methods Section 2.3**) for the selection of terminal exons containing tandem poly(A) sites with sufficient expression and read coverage to reliably infer the usage of the poly(A) sites. As 3’ end processing sites are relatively poorly annotated in the human genome (*Gruber et al., 2016, Genome Research*; PMID: 27382025), but MAPP depends on knowing the exact location of cleavage and polyadenylation sites for its impact maps, only poly(A) sites with experimental evidence from dedicated 3’ end sequencing protocols are taken into account. We have previously identified the vast majority of 3’ end sequencing datasets available within the public domain, performed rigorous quality control and developed a uniform processing pipeline to filter out background noise and integrate all high quality sites into a comprehensive resource that provides experimental evidence for the exact position of poly(A) sites (*Herrmann et al, 2021, Nucleic Acids Res*, PMID: 31617559). For the quantification of poly(A) site usage MAPP applies its “Poly(A) site Quantification” (short PAQ) module (**Figure 2.2d, below; Supplementary Methods Section 2.5**). For the quantification of poly(A) site usage we make use of our previously developed and thoroughly validated PAQR tool, which was recently benchmarked to be among the top performing tools for this kind of task (PMID: 37816550). While analysing every RNA-Seq sample separately PAQR takes into consideration per-site sufficient read coverage, i.e. the read coverage profile in the vicinity of the site. Sites that can not be reliably quantified based on the available mapped reads are marked for subsequent removal, i.e. before the statistical modelling. For further details please see **Supplementary Methods Section 2.5** and (*Gruber et al. 2018, Genome Biology*; PMID: 29592812).

Figure 2.2 (new Supplementary Figs: S20, S24, S22, S26): Snakemake rule graphs for the MAPP modules: (a) “Alternatively Spliced Exons” (ASE), (b) “Quantify Exon Inclusion” (QEI), (c) “Tandem Poly(A) sites” (TPA), (d) “PAQR” (PAQ). These modules are described in detail in **Supplementary Methods** sections: **2.2, 2.4, 2.3** and **2.5**, respectively.

3. The authors claimed that they did not select significant regulated AS or APA events to identify their regulatory motifs. Instead, they included all the database-annotated events which different regulatory levels to train the model. But, during most biological processes, only a small fraction of AS and APA events show significant regulation or have enough read coverage to reliably infer their expression levels. The general approach the authors proposed probably cannot be applied to many biological conditions.

We would like to thank the reviewer for bringing up how our approach deals with small sets of differentially processed splice sites and poly(A) sites, respectively. Indeed, many studies aiming to identify regulatory elements do so by comparing the sequence composition around/within regions of interest (genes/exons/etc.) that exhibit a significant change upon perturbations, with the composition around/within regions that do not show a significant change. Our approach is different in that it quantitatively models the change in all regions that are reliably quantified (please see the response to the previous question). Thus, rather than calculating enrichments from two classes of regions, we quantitatively model the change in terms of motif activities. Significant regulation, significant change in activity and reliability of expression estimates are distinct concepts. For the latter, we enforce stringent criteria for selecting reliably quantified regions. For the second, we quantitatively model the response in terms of motif activities (**Supplementary Methods Sections 2.7 and 2.8**). For inferring significant regulation we have implemented rigorous statistical tests.

Concerning the reviewers question about the performance of MAPP on datasets that exhibit only small numbers of differential regulated events, we can confirm that MAPP is able to also infer meaningful signal in such cases, as long as the occurrence of the changes is indeed linked to the occurrence of a specific sequence motif in the investigated regions (which can be adapted as required, see **Supplementary Methods Section 2.6**). That is, when running in k-mer mode, MAPP covers the entire sequence space and will thus be able to identify any sequence element that can quantitatively explain the observed changes, even if the numbers might be small. However, this is of course only true if it is the case that most regions that contain the sequence element respond coherently. If they do not, the sequence element does not explain the response on its own. This outcome would not be a limitation of MAPP, but it would simply make most sense, as the changes could obviously not be quantitatively explained by the presence / absence of the sequence element.

Please also note that MAPP infers for all tested TPM thresholds presented in **Figure 2.1** the same k-mer as being the most significant, which demonstrates that the MAPP inferences are stable in terms of varying event numbers due to the chosen TPM threshold (the higher the threshold, the smaller the number of considered events). For the three validation datasets using a TPM threshold of 1 or 15 as well as any value in between infers the correct binding motif from the data. Only the significance of the motif activity increases with lowering the threshold, as the statistical power of the model increases.

4. The software mainly outputs the meta plots of motifs and their significance scores around exons and polyA sites. It would be better if the software can output significantly regulated AS and APA events as well as the motif existence for each event, which will be helpful for experimental biologicals to find interesting genes to follow up.

We thank the reviewer for this suggestion, which we followed. In our revised version we vastly extended our software to also provide now to the user detailed information and corresponding downloads for all individual exons / poly(A) sites reliably quantified within each sample. In particular, we have redesigned our main report and implemented 7 novel download pages that contain descriptions of the data tables provided with every MAPP result (Figure 2.3, below). As the reviewer has pointed out, it will be very valuable for experimental biologists to have available the gene, transcript, exon and poly(A) site quantifications for all the samples analysed with MAPP as well as the binding sites MAPP is operating on. All of these data can now be accessed via our new MAPP report (Figure 2.3, below).

Figure 2.3: (a) Old MAPP report consisting of one single HTML page. (b) New MAPP report, now containing 7 novel information and download pages that provide all details on individual events that users might download for downstream validations and follow-up analyses.

5. The Method section tends to lack critical detailed information about the data analysis steps. They described the concept of the models but without any detailed information about data quality controls or parameters they used for the analyses of individual samples. For example, how did not they do the read mapping? How did they ensure the genes, exons or polyA sites are well expressed? It is hard to judge the rigor of their analyses.

We thank the reviewer for the suggestion to not only provide the code and corresponding documentation plus a short description of the MAPP modules in their respective **Supplementary Methods** sections, but to describe the modules in more detail and also outline and discuss the individual MAPP parameters the expert user might adjust for tailored MAPP runs. Accordingly, we have now added a complete list of parameters for every MAPP module plus a description of them and their used default value. In addition, for every MAPP module we have added a separate workflow graph, in order to provide an overview of the order in which all the software tools and scripts of each module are connected to each other.

Figure 2.4 (new Supplementary Fig. S18): Snakemake rulegraph for the “Preprocessing and Quality Analysis” module (short PQA).

Concerning the quality control the reviewer specifically asked about, below we are providing a brief description (more Details can be found in the **Supplementary Methods Section 2.1**):

- The raw RNA-seq reads are analysed by the **FastQC tool** which reports various RNA-seq specific quality scores, such as per-base sequence qualities, per-base N content, adapter content as well as GC-content.
- MAPP also makes use of the **RNA-SeQC package**, which provides detailed quality metrics regarding the genomic alignments. Among the most important ones are the total fraction of mapped reads, the fraction of unique reads mapped, the fraction of

high quality reads, the fraction of intergenic reads, and the fraction of rRNA-mapped reads.

- To have a measure of uniform read coverage and RNA degradation, MAPP calculates the TIN (Transcript Integrity Number) score.

By default MAPP determines for every sample all of the above mentioned quality metrics, thus providing an array of quality measures that can be investigated by the user (as required). Most importantly, MAPP enables to filter out low quality samples from a given dataset in a fully automatic manner. That is, when running MAPP with the “quality_check” parameter being set to ‘true’, which is the MAPP default (see **Supplementary Methods Section 2.1**), MAPP automatically removes all samples that do not pass all the quality parameters important for a high quality MAPP run. These MAPP quality parameters are set to a default value that guarantees optimal MAPP results. However, if required these parameters can be set by the user as described in **Supplementary Methods Section 2.1**.

Finally, we would like to thank the reviewer for pointing out that in our first version of the manuscript, we hadn’t yet integrated our GitHub modules documentation properly into our **Supplementary Methods** sections, which we have done now following the reviewer’s suggestion.

6. Another major concern I have is that the manuscript did not describe solid novel biological findings. The motifs associated with the factor knockdowns were limited to the factor itself. An advantage of the approach is to study the interplay between different RBPs. The authors may discover some novel interplay in the RBP knockdown datasets.

We very much appreciate the great suggestion by the reviewer to use our ENCODE screening results to provide novel insights into the potential interplay of specific RBPs.

Figure 2.5 (new Manuscript Fig. 3b): MAPP impact maps of selected HNRNPK (ENCSR853ZJS) and PCBP1 (ENCSR635FRH) RBP knock-down (KD) experiments. Colored bars indicate regions in which exclusively HNRNPK (green), exclusively PCBP1 (yellow), or both RBPs (red) regulate pre-mRNA processing by binding the ‘CCC’ motif.

Following the reviewer's suggestion, in the revised manuscript we have expanded our analysis of the ENCODE datasets, clustering RBPs by their MAPP 3' splice site (3'SS), 5' splice site (5'SS) and poly(A) site (PAS) results to group RBPs that are inferred to act on similar sequence motifs. We present now one selected example, HNRNPK and PCBP1, as a novel subfigure in our revised manuscript (**Figure 2.5 below**; revised **Manuscript Fig. 3b**), and report the other candidates in a comprehensive novel Supplementary Figure (**Figure 2.6, below**; **Supplementary Fig. S5**).

Figure 2.6 (new Supplementary Fig. S5): Clustering of the ENCODE top ranked MAPP results for 5' splice sites (5'SS), 3' splice sites (3'SS) and poly(A) sites (PAS) performed as described in the **Supplementary Methods Section 2.13**. K-mers which cluster together are marked with dashed boxes. Additionally, position weight matrices from the curated ATtRACT database which correspond to the targeted protein are reported next to the impact maps, where available and in decent accordance to the top ranked motif.

7. A novelty seems to be the finding of PTBP1 and RBFOX motifs in Glioblastoma. The results were limited to the correlation of gene expression and regulatory events. It is unclear whether PTBP1 and RBFOX as well as their interactions are indeed important to Glioblastoma.

As the reviewer states correctly, applying MAPP to normal and malignant brain tissue samples we observe that the motifs bound by the Polypyrimidine Tract Binding Protein 1 (PTBP1) and the RNA Binding Fox Homolog (RBFOX) RBPs coordinately malregulate splicing in glioblastomas, where hundreds of exon skipping events can be explained by the activities of these regulators (**Manuscript Figs. 4b,c**). That is, MAPP infers the sequence elements that can best explain the observed splicing changes. Importantly, in our manuscript we demonstrate that the set of exons having regulatory sites for both regulators has the largest changes in percent-spliced-in (**Manuscript Fig. 5a; Supplementary Fig. S10**) and also contains multiple cassette exon switching events that were previously shown experimentally to contribute to a more malignant cellular state (**Manuscript Fig. 5c**). We agree with the reviewer that these experimentally already validated connections were not yet properly highlighted and thus we have now improved the discussion of the 5 selected cassette exon candidates that we find to be significantly ($p < 0.05$) less included in GBM compared to normal brain in our samples and that were previously shown to contribute to the malignant properties of glioblastomas. Below we provide how this reads now in our revised manuscript for two examples (additional three candidates can be found on page 15 of our revised Manuscript):

- I. *“Exon 6 of the ANXA7 gene (Supplementary Table S7) skipped in GBM (Fig. 5c), provides an interesting positive control for our analysis. Inclusion of this exon was previously shown to be regulated by PTBP1 and skipping of ANXA7 exon 6 promotes the progression of GBM by fostering angiogenesis³⁵. Thus it provides an interesting link between the MAPP-inferred activities and the molecular properties of GBM. ANXA7 exon 6 has binding sites for the PTBP1 RBP in the regions that MAPP infers to be regulated by the PTBP1 RBP (Supplementary Table S7).”*
- II. *“Consistent with our observations in GBM (Fig. 5c), a previous study has shown that high levels of PTBP1 cause RTN4 exon 3 skipping. In contrast, PTBP1 knock-down results in enhanced inclusion of RTN4 exon 3. Importantly, the overexpression of the exon 3-including RTN4 splice isoform was shown to decrease the proliferation of glioma cell lines, whereas skipping of RTN4 exon 3, as observed in GBM (Fig. 5c), contributes significantly to their rapid growth characteristics³⁸.”*

Thus, we discuss experimental evidence for the importance of PTBP1-RBFOX co-regulated exons to more malignant cellular states. Furthermore, following the reviewer's recommendation to expand on the biological implications of our analysis, we have investigated from a splicing perspective the hypothesis of the field that the GBM's cells of origin might be neural progenitor cells (PMID: 37109434). We show that the activities of PTBP1 and the RBFOX RBPs are quite specific compared to other tissues (novel **Supplementary Figure S13**). Moreover, we have also applied MAPP to infer the splicing activities for neural progenitor cells and differentiated neurons.

Interestingly, we find that in terms of the expression levels and splicing activities of the PTBP1 and RBFOX RBPs, GBM cells are much more similar to neural progenitors than to differentiated neurons. We present our results in an entirely new main figure, consisting of 9 subfigures, in our revised manuscript (**Figure 2.7 below; Manuscript Fig. 6**).

Figure 2.7 (entirely new **Manuscript Fig. 6**) | **Regulation of cassette exon switching in neuronal differentiation and glioblastomas.** **a** | Scatter plots of the PTBP RBPs mRNA expression levels versus the activities of the PTBP-associated CUCU motif for the indicated region window as inferred by MAPP for cells from the ventricular zone (n=6) and cells from the cortical plate (n=6). R indicates the Pearson correlation coefficient and p the corresponding p-value. The dashed gray line shows the linear regression. **b** | MAPP inferred impact maps of the PTBP-associated CUCU motif at 3' splice sites (3' SS) and the RBFOX-associated UGCAUG motif at 5' splice sites (5' SS) for the two cell populations described in (a). **c** | Scatter plots as in (a) but for the RBFOX RBPs mRNA expression levels versus the activities of the RBFOX-associated UGCAUG motif for the indicated region window as inferred by MAPP for the cell populations described in (a). **d** | Scatter plots as in (a) but for a neural stem cell model (n=9) and normal brain samples (n=9). **e** | MAPP inferred impact maps as in (b) but for the cells described in (d). **f** | Scatter plots as in (c) but for the cells described in (d). **g** | Distribution of differences between percent-spliced-in averages (ΔPSI) observed in the cell populations described in (a) (left panel) and for the cells described in (d) (right panel) for cassette exons regulated by PTBP1 and/or RBFOX according to having a binding probability c of ≥ 0.9 within the regions inferred by MAPP to be significantly regulated. The number of cases of each group is indicated. **h** | Comparison of the motif activities (in the windows indicated in a,c,d,f) and the expression levels of the

PTBP1 and RBFOX RBPs in cells residing in the ventricular zone (VZ; n=6) and the cortical plate (CP; n=6) (top panel) to those observed in glioblastoma (GBM; n=5) and normal brain (NORMAL; n=5) (bottom panel). Wide boxes in the violin plots indicate the interquartile range (IQR) with the white dot inside corresponding to the median, narrow lines indicating 1.5 times the IQR from the hinge. Individual data points are shown as colored dots (x). **i** | Graphical summary of cassette exon co-regulation by the PTBP1 and RBFOX RBPs in the indicated cell systems.

8. For the analyses of glioblastoma samples, they listed 5 healthy and 5 tumor samples in the supplementary tables. It is unclear how these 10 samples were selected from the hundreds of clinically available samples. Will the authors' conclusions still be valid if they apply their analyses to all samples?

Indeed the reviewer is correct in pointing out that in the initially submitted version of our manuscript our analysis of healthy and tumor samples was based on only 5 healthy and 5 glioblastoma samples. In our revised manuscript we show that our observations are replicated in two other glioblastoma cohorts and are thus general to the disease. In particular, we present now novel MAPP results for 30 additional glioblastoma and normal samples (**Figure 2.8 below**; now also **Supplementary Fig. S9**) as well as for another 98 glioblastoma and normal brain samples (**Figure 2.9 below**, now also **Supplementary Fig. S8**), thereby extending our study by two more independent cohorts and increasing our glioblastoma sample size from 5 in our initial version of the manuscript to 104 glioblastoma samples in total (all samples are listed in **Supplementary Table S1**).

Figure 2.8 (new Supplementary Fig. S9): (a) MAPP results for normal brain, glioblastoma, oligodendrogloma and astrocytoma samples (Dataset accessions: GSE147352, GSE185861 (two runs for each donor)) for the PTBP-bound CTCT k-mer (left panel) as well as for the RBFOX-bound GCATG k-mer (right panel). Region definitions for the MAPP windows are shown on top of each impact map. MAPP was run in k-mer mode (covering k-mer length 3-5). There was no constraint for a minimum exon length in order to also account for micro-exons in neurons. (b) Scatter plots of the MAPP inferred activities for the indicated region windows versus the PTBP and RBFOX RBP mRNA expression levels observed for each sample.

Figure 2.9 (new Supplementary Fig. S8): (a) MAPP results for normal brain (NORMAL) and glioblastoma (GBM) samples (Dataset accession: GSE147352) for the PTBP-bound CTCT k-mer (left panel) as well as for the RBFOX-bound GCATG k-mer (right panel). The exact region definitions for each window are indicated on top of both impact maps. MAPP was run in k-mer mode (covering k-mer length 3-5) and without a minimum exon length constraint in order to also account for micro-exons prevalent in neurons. (b) Scatter plots of the MAPP inferred activities for the region windows indicated versus the PTBP and RBFOX RBP mRNA expression levels.

In summary, to date MAPP is the only modelling approach able to identify regulators that explain genome-scale changes in both splicing and polyadenylation from standard RNA sequencing data in an integrated manner. Also, we have developed the splicing model from scratch, further developed and greatly improved the 3' end processing model and integrated both approaches using novel statistical approaches towards obtaining a quantitative view on pre-mRNA processing. Thus, MAPP addresses the great need for a method that identifies key regulatory elements of pre-mRNA processing from standard RNA-seq data and opens the door for an abundance of studies that make use of RNA-seq data of RBP perturbation experiments or physiological conditions, such as cancers.

We show that MAPP is able to infer known and novel interactions, and by applying it to normal and malignant brain samples we demonstrate that MAPP is capable of inferring key pre-mRNA processing regulators from complex physiological conditions.

Finally, we would like to wholeheartedly thank the reviewer once again for all the valuable input and suggestions and also for carefully reading and considering our vastly extended analysis and revised manuscript. Your valuable comments aided us in strengthening scientific rigour of this important work of our still young research group. Many thanks!

REVIEWERS' COMMENTS

Reviewer #1 (Remarks to the Author):

The authors have addressed all my concerns.

Reviewer #2 (Remarks to the Author):

The authors did a good job addressing my comments. I recommend this work to be published.

Reviewer #2 (Remarks on code availability):

The codes and instructions were clearly written.